# Denoising Diffusion Variational Inference

## Abstract

Latent variable methods are a powerful tool for representation learning that greatly benefit from expressive variational posteriors, including generative models based on normalizing flows or adversarial networks. In this work, we propose denoising diffusion variational inference, which relies on diffusion models—recent generative algorithms with state-of-the-art sample quality—to fit a complex posterior by performing diffusion in latent space. Our method augments a variational posterior with auxiliary latent variables via a user-specified noising process that transforms a complex latent into a simple auxiliary latent. The approximate posterior then reverses this noising process by optimizing a lower bound on the marginal likelihood inspired by the wake-sleep algorithm. Our method can be used to fit deep latent variable models, which yields the DiffVAE algorithm. This algorithm is especially effective at dimensionality reduction and representation learning, where it outperforms methods based on adversarial training or invertible flow-based posteriors. We use this algorithm on a motivating task in biology—inferring latent ancestry from human genomes—and show that it outperforms strong baselines on the 1000 Genomes dataset.

## 1 Introduction

Latent variables are a powerful tool for learning representations in both machine learning (Kingma & Welling, 2013) and its applications, including in fields of science such as biology (Battey et al., 2021). Latent variable methods often rely on variational inference to fit an approximate model of the posterior distribution (Vahdat & Kautz, 2020; Maaløe et al., 2016). The expressivity of this model has a significant impact on the performance of variational inference (Kingma et al., 2016), which motivates research that leverages modern generative models—including normalizing flows (Rezende & Mohamed, 2015) and generative adversarial networks (Goodfellow et al., 2014)—to represent expressive approximate posteriors.

This work seeks to improve variational inference via expressive posteriors based on diffusion-based algorithms, a modern class of generative models (Ho et al., 2020; Nichol & Dhariwal, 2021; Song et al., 2020). Diffusion methods are defined via a noising process, which maps data into Gaussian noise; a diffusion model generates data by reversing this noising process, which yields very realistic and high-quality samples. Here, we argue for using diffusion models in latent space, where we gradually map a simple (e.g., Gaussian) latent representation of the data into one that is more complex via an iterative diffusion-like procedure. This procedure yields an expressive approximate posterior trained with a denoising objective that does not involve adversarial training (Makhzani et al., 2015) or constrained invertible architectures (Kingma et al., 2016).

Specifically, we propose denoising diffusion variational inference, an approximate inference algorithm that introduces auxiliary latent variables into either the model or the approximate posterior via a user-specified noising process. This process transforms the latent variable we seek to model into a simple (e.g., Gaussian) auxiliary latent; during inference, we fit the approximate posterior by reversing this noising process. Our learning objective is a variational lower bound inspired by the wake-sleep algorithm (Hinton et al., 1995) that can be interpreted as a form of regularized variational inference. We also derive extensions of our method to semi-supervised learning and clustering.

Our method can naturally fit deep latent variable models, in which case we denote the resulting algorithm as a variational autoencoder with a diffusion posterior (DiffVAE). We find that DiffVAEs are most effective on tasks in dimensionality reduction and visualization, especially when compared

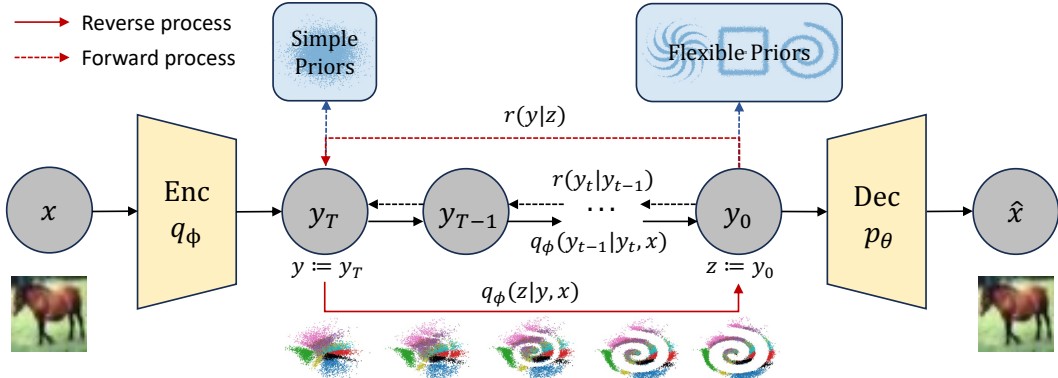

Figure 1: Flow chart demonstrating denoising diffusion variational inference. Between the encoder and decoder, we have a diffusion model to map a simple prior into any complex prior.

against alternative families of approximate posteriors, such as adversarial autoencoders (Makhzani et al., 2015). We also evaluate DiffVAEs on the MNIST and CIFAR-10 datasets and on a real problem in biological data analysis—inferring human ancestry from genetic data. Our method outperforms strong baselines on the 1000Genomes (Siva, 2008) and maps data into a low dimensional space in a way that preserves semantically meaningful structure (Haghverdi et al., 2015).

**Contributions.** In summary, this work introduces denoising diffusion variational inference, an approximate inference algorithm that features three key components: auxiliary latent variables, a user-specified noising process over these variables, and a lower bound on the marginal likelihood inspired by wake-sleep. Our method can be used to fit deep latent variable models, which yields the DiffVAE algorithm. This algorithm is especially effective at dimensionality reduciton and representation learning, where it outperforms alternative methods based on adversarial training.

## 2 BACKGROUND

**Deep Latent Variable Models** Latent variable models (LVMs) $p_{\boldsymbol{\theta}}(\mathbf{x}, \mathbf{z})$ are usually fit by optimizing the evidence lower bound (ELBO) $\log p_{\boldsymbol{\theta}}(\mathbf{x}) \leq \mathbb{E}_{q_{\boldsymbol{\phi}}(\mathbf{z}|\mathbf{x})}[\log p_{\boldsymbol{\theta}}(\mathbf{x}|\mathbf{z})] - D_{\mathrm{KL}}(q_{\boldsymbol{\phi}}(\mathbf{z}|\mathbf{x})||p_{\boldsymbol{\theta}}(\mathbf{z}))$, which serves as a tractable surrogate for the marginal log-likelihood (MLL). The gap between the MLL and the ELBO equals precisely $D_{\mathrm{KL}}(q_{\boldsymbol{\phi}}(\mathbf{z}|\mathbf{x})||p_{\boldsymbol{\theta}}(\mathbf{z}|\mathbf{x}))$—thus, a more expressive $q_{\boldsymbol{\phi}}(\mathbf{z}|\mathbf{x})$ may better fit the true posterior and induce a tighter ELBO (Kingma & Welling, 2013).

Expressive variational posteriors can be formed by choosing more expressive model families—including auxiliary variable methods (Maaløe et al., 2016), MCMC-based methods (Salimans et al., 2015), normalizing flows (Rezende & Mohamed, 2015)—or improved learning objectives—e.g., adversarial or sample-based losses (Makhzani et al., 2015; Zhao et al., 2017; Si et al., 2022; 2023).

The wake-sleep algorithm (Hinton et al., 1995) optimizes an alternative objective $\mathbb{E}_{q_{\boldsymbol{\phi}}(\mathbf{z}|\mathbf{x})}[\log p_{\boldsymbol{\theta}}(\mathbf{x}|\mathbf{z})] - D_{\mathrm{KL}}(p_{\boldsymbol{\theta}}(\mathbf{z}|\mathbf{x})||q_{\boldsymbol{\phi}}(\mathbf{z}|\mathbf{x}))$, in which the KL divergence term is reversed. The learning procedure for wake-sleep involves alternating between "wake" phases where the recognition model is updated and "sleep" phases where the generative model is refined.

**Denoising Diffusion Models** A diffusion model is defined via a user-specified noising process $q$ that maps data $\mathbf{x}_0$ into a sequence of $T$ variables $\mathbf{y}_{1:T} = \mathbf{y}_1, ..., \mathbf{y}_T$ that represent increasing levels of corruption to $\mathbf{x}_0$. We obtain $\mathbf{y}_{1:T}$ by applying a Markov chain $q(\mathbf{y}_{1:T}|\mathbf{x}_0) = \prod_{t=1}^{T} q(\mathbf{y}_t|\mathbf{y}_{t-1})$, where we define $\mathbf{y}_0 = \mathbf{x}_0$ for convenience. When $\mathbf{x}_0$ is a continuous vector, a standard choice of transition kernel is $q(\mathbf{x}_t \mid \mathbf{x}_{t-1}) = \mathcal{N}(\mathbf{y}_t; \sqrt{\boldsymbol{\alpha}_t}\mathbf{y}_{t-1}, \sqrt{1 - \boldsymbol{\alpha}_t}\mathbf{I})$, which is a Gaussian centered around a copy of $\mathbf{y}_{t-1}$ to which we added noise following a schedule $0 < \boldsymbol{\alpha}_1 < \boldsymbol{\alpha}_2 < ... < \boldsymbol{\alpha}_T = 1$.

A diffusion model can then be represented as a latent variable distribution $p(\mathbf{x}_0, \mathbf{y}_{1:T})$ that factorizes as $p(\mathbf{x}_0, \mathbf{y}_{1:T}) = p(\mathbf{y}_T) \prod_{t=0}^{T-1} p_{\boldsymbol{\theta}}(\mathbf{y}_t \mid \mathbf{y}_{t+1})$ (again using $\mathbf{y}_0$ as shorthand for $\mathbf{x}_0$). This model seeks to approximate the reverse of the forward diffusion $q$ and map noise $\mathbf{y}_T$ into data $\mathbf{x}_0$.

The true reverse of the process $q$ cannot be expressed in closed form; as such, we parameterize $p_{\boldsymbol{\theta}}$ with parameters $\boldsymbol{\theta}$ trained by maximizing the ELBO: $\log p_{\boldsymbol{\theta}}(\mathbf{x}_0) \geq \mathbb{E}_q \left[ \log p_{\boldsymbol{\theta}}(\mathbf{x}_0|\mathbf{x}_1) - \sum_{t=2}^T D_{\mathrm{KL}}(q(\mathbf{x}_{t-1}|\mathbf{x}_t, \mathbf{x}_0) || p_{\boldsymbol{\theta}}(\mathbf{x}_{t-1}|\mathbf{x}_t)) \right] - D_{\mathrm{KL}}(q(\mathbf{x}_T|\mathbf{x}_0) || p(\mathbf{x}_T))$

**Visualization and Dimensionality Reduction**  There exist two important types of dimensionality reduction methods, which (1) emphasize the preservation of pairwise distance structures among all data samples, e.g., PCA (Wold et al., 1987) and LDA (Balakrishnama & Ganapathiraju, 1998), and (2) local distances over global ones, e.g., t-SNE (Van der Maaten & Hinton, 2008) and UMAP (McInnes et al., 2018). Latent variable models (LVMs) (Kingma et al., 2016; Makhzani et al., 2015) are another class of techniques for dimensionality reduction. They represent high-dimensional data in terms of latent, or hidden, variables in a lower-dimensional space, effectively providing a compact representation of the data.

## 3  VARIATIONAL INFERENCE WITH DENOISING DIFFUSION MODELS

We introduce *denoising variational inference*, which enhances variational inference with diffusion-based methods and is motivated by challenges in data visualization and dimensionality reduction. Our approach consists in augmenting variational inference in a latent variable model $p(\mathbf{x}, \mathbf{z})$ with *auxiliary latents* $\mathbf{y} \in \mathcal{Y}$ introduced via a user-specified *noising process* $r(\mathbf{y}|\mathbf{z})$. The $r(\mathbf{y}|\mathbf{z})$ transforms $\mathbf{z}$—which is the latent whose intractable posterior we seek to approximate—into $\mathbf{y}$, whose posterior will be easier to model. Examples of $r$ include forward diffusion processes, discrete noising processes (Austin et al., 2021), as well as custom regularizers (Section A.2).

We then form an expressive posterior $q(\mathbf{z}|\mathbf{x})$ by fitting the reverse of the noising process $r(\mathbf{y}|\mathbf{z})$ as in a diffusion model. We define $q(\mathbf{z}|\mathbf{x})$ by sampling from a first model $q(\mathbf{y}|\mathbf{x})$—this is an easier task since we can choose $y$ to have a simple (e.g., Gaussian) posterior—and then by sampling from a denoising model $q(\mathbf{z}|\mathbf{x}, \mathbf{y})$ that approximates the reverse process $r(\mathbf{z}|\mathbf{y})$. The model $q(\mathbf{z}|\mathbf{x}, \mathbf{y})$ is parameterized and trained as a denoising diffusion model, and is thus a flexible posterior estimator.

When diffusion variational inference is used to fit a variational autoencoder, we refer to the resulting algorithm as VAE with diffusion encoders (DiffVAE). We define the full algorithm below.

### 3.1  DIFFVAE: VARIATIONAL AUTOENCODERS WITH DIFFUSION ENCODERS

We seek to fit a latent variable model $p_{\boldsymbol{\theta}}(\mathbf{x}, \mathbf{z})$ with a potentially complex prior $p_{\boldsymbol{\theta}}(\mathbf{z})$. One perspective that can be used to define our approach consists in *lifting* $p_{\boldsymbol{\theta}}(\mathbf{x}, \mathbf{z})$ into an extended latent space by introducing auxiliary latents $\mathbf{y} \in \mathcal{Y}$ and applying variational inference in the extended space.

Specifically, we augment the model with the aforementioned user-specified noising process $r(\mathbf{y}|\mathbf{z})$. We use $p_{\boldsymbol{\theta}}(\mathbf{x}, \mathbf{y}, \mathbf{z}) = r(\mathbf{y}|\mathbf{z})p_{\boldsymbol{\theta}}(\mathbf{x}, \mathbf{z})$ to denote the extended probability distribution. Note that the marginalizing out $\mathbf{y}$ in $p_{\boldsymbol{\theta}}(\mathbf{x}, \mathbf{y}, \mathbf{z})$ yields the original model: hence fitting $p_{\boldsymbol{\theta}}(\mathbf{x}, \mathbf{y}, \mathbf{z})$ in an extended probability space solves our original task.

The noising process $r$ may also introduce multiple latents $\mathbf{y}_{1:T}$, as in the forward process $r(\mathbf{y}_{1:T}|\mathbf{z}) = \prod_{t=1}^{T-1} r(\mathbf{y}_{t+1}|\mathbf{y}_t, \mathbf{z})$ of a diffusion model; we may then define $r(\mathbf{y}|\mathbf{z}) = r(\mathbf{y}_T|\mathbf{z}) := \int r(\mathbf{y}_T, \mathbf{y}_{1:T-1}|\mathbf{z}) d\mathbf{y}_{1:T-1}$. While we do not require this specific form for $r$, we will return to diffusion noising processes later in the section.

#### 3.1.1  DENOISING VARIATIONAL INFERENCE

One possible way to apply variational inference to $p_{\boldsymbol{\theta}}(\mathbf{x}, \mathbf{y}, \mathbf{z})$ is to apply the ELBO twice to obtain:

$$\log p_{\boldsymbol{\theta}}(\mathbf{x}) \geq \mathbb{E}_{q_{\boldsymbol{\phi}}(\mathbf{y}|\mathbf{x})}[\log p_{\boldsymbol{\theta}}(\mathbf{x}|\mathbf{y})] - D_{\mathrm{KL}}(q_{\boldsymbol{\phi}}(\mathbf{y}|\mathbf{x}) || p_{\boldsymbol{\theta}}(\mathbf{y})) \tag{1}$$

$$\geq \mathbb{E}_{q_{\boldsymbol{\phi}}(\mathbf{y}, \mathbf{z}|\mathbf{x})}[\log p_{\boldsymbol{\theta}}(\mathbf{x}|\mathbf{z}) - D_{\mathrm{KL}}(q_{\boldsymbol{\phi}}(\mathbf{z}|\mathbf{x}, \mathbf{y}) || p_{\boldsymbol{\theta}}(\mathbf{z}|\mathbf{y}))] - D_{\mathrm{KL}}(q_{\boldsymbol{\phi}}(\mathbf{y}|\mathbf{x}) || p_{\boldsymbol{\theta}}(\mathbf{y})) \tag{2}$$

While Equation 2 is a valid learning objective, it does not yield a training procedure comparable to that of a diffusion model. Diffusion model training involves sampling noisy data from the forward process; here, we sample from the approximate reverse process (from $q_{\boldsymbol{\phi}}(\mathbf{z}|\mathbf{y})$) and seek to match the true process $r(\mathbf{z}|\mathbf{y})$. In practice, we have found the learning signal from this procedure to be too weak to learn a good $q_{\boldsymbol{\phi}}(\mathbf{z}|\mathbf{y})$ that reverses the noising process $r(\mathbf{y}|\mathbf{z})$.

Instead, we adopt a learning objective $\mathcal{L}(\mathbf{x}, \boldsymbol{\theta}, \boldsymbol{\phi})$ inspired by the wake-sleep algorithm:

$$\mathcal{L} = \underbrace{\mathbb{E}_{q_{\boldsymbol{\phi}}(\mathbf{y},\mathbf{z}|\mathbf{x})}[\log p_{\boldsymbol{\theta}}(\mathbf{x}|\mathbf{z})]}_{\text{wake / recons. term } \mathcal{L}_{\text{rec}}(\mathbf{x},\boldsymbol{\theta},\boldsymbol{\phi})} - \underbrace{D_{\text{KL}}(q_{\boldsymbol{\phi}}(\mathbf{y},\mathbf{z}|\mathbf{x})|p_{\boldsymbol{\theta}}(\mathbf{y},\mathbf{z}))}_{\text{prior regularization term } \mathcal{L}_{\text{reg}}(\mathbf{x},\boldsymbol{\theta},\boldsymbol{\phi})} - \underbrace{\mathbb{E}_{p_{\boldsymbol{\theta}}(\mathbf{x})}[D_{\text{KL}}(p_{\boldsymbol{\theta}}(\mathbf{z}|\mathbf{x})||q_{\boldsymbol{\phi}}(\mathbf{z}|\mathbf{x}))]}_{\text{sleep term } \mathcal{L}_{\text{sleep}}(\boldsymbol{\phi})} \tag{3}$$

Observe that this objective is the ELBO in Equation (2) augmented with an additional regularizer $\mathcal{L}_{\text{sleep}}(\boldsymbol{\phi})$ that is inspired by the sleep phase of the wake-sleep algorithm. This term consists of the reverse KL divergence $D_{\text{KL}}(p_{\boldsymbol{\theta}}(\mathbf{z}|\mathbf{x})|q_{\boldsymbol{\phi}}(\mathbf{z}|\mathbf{x}))$. Section 3.1.2 shows that this divergence can be optimized via a diffusion-like training procedure. As in wake-sleep, we optimize $\mathcal{L}_{\text{sleep}}$ over $\boldsymbol{\phi}$ only.

Objective (7) poses restrictions on $p_{\boldsymbol{\theta}}$ and $r$. The prior $p_{\boldsymbol{\theta}}(\mathbf{y}, \mathbf{z})$ needs to have a tractable density, although we will also present approximations and experimental results with implicit sample-based priors. Additionally, $q_{\boldsymbol{\phi}}(\mathbf{y}, \mathbf{z}|\mathbf{x})$ must feature tractable entropy and sampling.

Lastly, note that $\mathcal{L}$ lower bounds the marginal log-likelihood $\log p_{\boldsymbol{\theta}}(\mathbf{x})$. When $q_{\boldsymbol{\phi}}(\mathbf{y}, \mathbf{z}|\mathbf{x})$ equals the true posterior, this bound is tight, since the ELBO is tight, and the sleep term also equals zero.

### 3.1.2 OPTIMIZATION USING WAKE-SLEEP IN LATENT SPACE

Next, we introduce optimization algorithms for diffusion variational inference. Maximizing $\mathcal{L}(x, \boldsymbol{\theta}, \boldsymbol{\phi})$ involves optimizing the sleep term $\mathcal{L}_{\text{sleep}}(\boldsymbol{\phi})$. This optimization is tractable since:

$$\mathcal{L}_{\text{sleep}}(\boldsymbol{\phi}) = -\mathbb{E}_{p_{\boldsymbol{\theta}}(\mathbf{x})}[D_{\text{KL}}(p_{\boldsymbol{\theta}}(\mathbf{z}|\mathbf{x})|q_{\boldsymbol{\phi}}(\mathbf{z}|\mathbf{x}))] = \mathbb{E}_{p(\mathbf{x},\mathbf{z})}[\log[q_{\boldsymbol{\phi}}(\mathbf{z}|\mathbf{x})/p_{\boldsymbol{\theta}}(\mathbf{z}|\mathbf{x})]] \tag{4}$$

$$= \mathbb{E}_{p_{\boldsymbol{\theta}}(\mathbf{z})p_{\boldsymbol{\theta}}(\mathbf{x}|\mathbf{z})}[\log q_{\boldsymbol{\phi}}(\mathbf{z}|\mathbf{x})] + \bar{H}(p_{\boldsymbol{\theta}}) \tag{5}$$

$$\geq \mathbb{E}_{p_{\boldsymbol{\theta}}(\mathbf{z})p_{\boldsymbol{\theta}}(\mathbf{x}|\mathbf{z})}[\mathbb{E}_{r(\mathbf{y}|\mathbf{z})}[\log(q_{\boldsymbol{\phi}}(\mathbf{y},\mathbf{z}|\mathbf{x})/r(\mathbf{y}|\mathbf{z}))]] + \bar{H}(p_{\boldsymbol{\theta}}) \tag{6}$$

where in Equation (6) we applied the ELBO with $r(\mathbf{y}|\mathbf{z})$ as our choice of variational posterior over the latent $\mathbf{y}$ in the distribution $q_{\boldsymbol{\phi}}$; $\bar{H}(p_{\boldsymbol{\theta}})$ is the expected conditional entropy of $p_{\boldsymbol{\theta}}(\mathbf{z}|\mathbf{x})$, a constant that does not depend on $\boldsymbol{\phi}$. By maximizing Equation (6), we fit $q(\mathbf{z}|\mathbf{x})$ to $p_{\boldsymbol{\theta}}(\mathbf{z}|\mathbf{x})$ while also fitting $q(\mathbf{z}|\mathbf{y})$ to the true reverse denoising process $r(\mathbf{z}|\mathbf{y})$ via the reconstruction term $\mathbb{E}_{r(\mathbf{y}|\mathbf{z})}[\log q_{\boldsymbol{\phi}}(\mathbf{z}|\mathbf{x}, \mathbf{y})]$, which is one of the terms comprising Equation (6). To fit the objective $\mathcal{L}(\mathbf{x}, \boldsymbol{\theta}, \boldsymbol{\phi})$, we propose introducing an additional sleep step in which we sample from $p_{\boldsymbol{\theta}}(\mathbf{x}, \mathbf{y}, \mathbf{z})$ in order to compute gradients for $\mathcal{L}_{\text{sleep}}(\boldsymbol{\phi})$.

This procedure mirrors wake-sleep; however, sampling $\mathbf{x}$ from $p_{\boldsymbol{\theta}}$ to obtain gradients for the sleep term introduces computational overhead. To address this issue, we propose *wake-sleep in latent space*, an algorithm that optimizes an approximation $\hat{\mathcal{L}}(\mathbf{x}, \boldsymbol{\theta}, \boldsymbol{\phi})$ of $\mathcal{L}$:

$$\hat{\mathcal{L}}(\mathbf{x}, \boldsymbol{\theta}, \boldsymbol{\phi}) = \underbrace{\mathbb{E}_{q_{\boldsymbol{\phi}}(\mathbf{y},\mathbf{z}|\mathbf{x})}[\log p_{\boldsymbol{\theta}}(\mathbf{x}|\mathbf{z})]}_{\text{wake / reconstr. term } \mathcal{L}_{\text{rec}}(\mathbf{x},\boldsymbol{\theta},\boldsymbol{\phi})} - \underbrace{D_{\text{KL}}(q_{\boldsymbol{\phi}}(\mathbf{y},\mathbf{z}|\mathbf{x})||p_{\boldsymbol{\theta}}(\mathbf{y},\mathbf{z}))}_{\text{prior regularization term } \mathcal{L}_{\text{reg}}(\mathbf{x},\boldsymbol{\theta},\boldsymbol{\phi})} - \underbrace{D_{\text{KL}}(p_{\boldsymbol{\theta}}(\mathbf{z})||q_{\boldsymbol{\phi}}(\mathbf{z}|\mathbf{x}))}_{\text{latent sleep term } \mathcal{L}_{\text{sleep}}(\mathbf{x},\boldsymbol{\phi})}. \tag{7}$$

We have replaced $\mathcal{L}_{\text{sleep}}(\boldsymbol{\phi})$ with a latent sleep term $\mathcal{L}_{\text{sleep}}(\mathbf{x}, \boldsymbol{\phi})$, in which $\mathbf{x}$ is given, and we only seek to fit the true reverse noising process $r(\mathbf{z}|\mathbf{y})$ independently of $\mathbf{x}$. We can similarly show that

$$\mathcal{L}_{\text{sleep}}(\mathbf{x}, \boldsymbol{\phi}) = \mathbb{E}_{p_{\boldsymbol{\theta}}(\mathbf{z})}[\log q_{\boldsymbol{\phi}}(\mathbf{z}|\mathbf{x})] + \bar{H}(p_{\boldsymbol{\theta}}) \geq \mathbb{E}_{p_{\boldsymbol{\theta}}(\mathbf{z})r(\mathbf{y}|\mathbf{z})}[\log(q_{\boldsymbol{\phi}}(\mathbf{y},\mathbf{z}|\mathbf{x})/r(\mathbf{y}|\mathbf{z}))] + \bar{H}(p_{\boldsymbol{\theta}}) \tag{8}$$

$$= -\mathbb{E}_{p_{\boldsymbol{\theta}}(\mathbf{z})}[D_{\text{KL}}(r(\mathbf{y}|\mathbf{z})||q_{\boldsymbol{\phi}}(\mathbf{y}|\mathbf{z},\mathbf{x}))] - D_{\text{KL}}(p_{\boldsymbol{\theta}}(\mathbf{z})||q(\mathbf{z}|\mathbf{x})), \tag{9}$$

where $\bar{H}(p_{\boldsymbol{\theta}})$ is an entropy term constant in $\boldsymbol{\phi}$. Thus, we minimize the forward KL divergence by sampling $\mathbf{z}$, and applying the noising process to get $\mathbf{y}$; the $q_{\boldsymbol{\phi}}$ is fit to denoise $\mathbf{z}$ from $\mathbf{y}$ as in Equation (6).

We optimize our bound on $\hat{\mathcal{L}}(x, \boldsymbol{\theta}, \boldsymbol{\phi})$ end-to-end using minibatch gradient descent over $\boldsymbol{\theta}, \boldsymbol{\phi}$. While the wake term is a reconstruction loss as in wake-sleep, the sleep term generates latent samples $\mathbf{z}, \mathbf{y}$ from $r(\mathbf{y}|\mathbf{z})p_{\boldsymbol{\theta}}(\mathbf{z})$ (by analogy with $p_{\boldsymbol{\theta}}(\mathbf{x}|\mathbf{z})p_{\boldsymbol{\theta}}(\mathbf{z})$ in normal wake-sleep); the denoiser $q_{\boldsymbol{\phi}}$ is trained to recover $\mathbf{z}$ from $\mathbf{y}$. Thus, we perform wake-sleep *in latent space*, which obviates the need for alternating wake and sleep phases, and allows efficient end-to-end training. A limitation of this approximation is that the sleep term does not fit $q_{\boldsymbol{\phi}}$ to the true $p_{\boldsymbol{\theta}}(\mathbf{z}|\mathbf{x}, \mathbf{y})$, and as a consequence $\hat{L}$ is not a tight lower bound on $\log p_{\boldsymbol{\theta}}(\mathbf{x})$. We may think of $\mathcal{L}_{\text{sleep}}(\mathbf{x}, \boldsymbol{\phi})$ as a regularizer to the ELBO.

### 3.1.3 DENOISING DIFFUSION VI WITH DIFFUSION-BASED ENCODERS

Our framework is naturally instantiated with diffusion models. Let $\mathbf{y}_0 = \mathbf{z}$ and $\mathbf{y}_T = \mathbf{y}$. The forward noising model can be defined as $r(\mathbf{y}_{1:T}|\mathbf{z}) = \prod_{t=1}^{T} r(\mathbf{y}_t|\mathbf{y}_{t-1})$, where the $\mathbf{y}_{0:T}$ are an extended set of latent variables that represent increasingly noised versions of $\mathbf{y}_0$. We parameterize the approximate reverse diffusion process as $q_\phi(\mathbf{y}_{0:T}|\mathbf{x}) = q(\mathbf{y}_T|\mathbf{x}) \prod_{t=1}^{T} q_\phi(\mathbf{y}_{t-1}|\mathbf{y}_t, \mathbf{x})$.

We can form a lower bound $\mathcal{L}_{\text{diff}}(\mathbf{x}, \phi)$ on the sleep term $\mathbb{E}_{p(\mathbf{z})} \log q_\phi(\mathbf{z}|\mathbf{x})$ in Equation (8) (where $q_\phi(\mathbf{z}|\mathbf{x}) = \int q_\phi(\mathbf{z}, \mathbf{y}_{1:T}|\mathbf{x}) d\mathbf{y}_{1:T}$) using the ELBO for a diffusion model:

$$\mathcal{L}_{\text{diff}} = \mathbb{E}_r \left[ \log q_\phi(\mathbf{z}|\mathbf{y}_1, \mathbf{x}) - \sum_{t=2}^{T} D_{\text{KL}}(r(\mathbf{y}_{t-1}|\mathbf{y}_t, \mathbf{y}_0)|q_\phi(\mathbf{y}_{t-1}|\mathbf{y}_t, \mathbf{x})) \right] - D_{\text{KL}}(r(\mathbf{y}_T|\mathbf{z})||q_\phi(\mathbf{y}|\mathbf{x})). \tag{10}$$

This bound is an instantiation of Equation (8) when $r$ is a diffusion process. Many choices of $r, q$ fit this framework. A common type of noising process is Gaussian diffusion, where we define $r(\mathbf{y}_t|\mathbf{y}_{t-1}) = \mathcal{N}(\mathbf{y}_t; \sqrt{1-\boldsymbol{\alpha}_t}\mathbf{y}_{t-1}, \boldsymbol{\alpha}_t \mathbf{I})$ for a suitable schedule $(\boldsymbol{\alpha}_t)_{t=1}^{T}$. We then adopt the parameterization $q_\phi(\mathbf{y}_{t-1}|\mathbf{y}_t, \mathbf{x}) = \mathcal{N}(\mathbf{y}_{t-1}; \mu_\phi(\mathbf{y}_t, \mathbf{x}, t), \Sigma_\phi(\mathbf{y}_t, \mathbf{x}, t))$. It is then common to parameterize $q_\phi$ with a noise prediction network $\boldsymbol{\epsilon}_\phi$ (Ho et al., 2020); the sum of KL divergences can be approximated by $E_{t, \boldsymbol{\epsilon}_t \sim r(\mathbf{y}_0, t)} ||\boldsymbol{\epsilon}_t - \boldsymbol{\epsilon}_\phi(\sqrt{\bar{\boldsymbol{\alpha}}_t}\mathbf{y}_0 + \sqrt{1-\bar{\boldsymbol{\alpha}}_t}\boldsymbol{\epsilon}_t, \mathbf{x}, t)||^2$. By our earlier argument, this objective encourages $q_\phi(\mathbf{y}_{0:T}|\mathbf{x})$ to match $r(\mathbf{y}_{0:T})$ as well as the true posterior $p(\mathbf{z}|\mathbf{x})$.

Lastly, we need to show that this choice of $q$ admits a tractable entropy. This follows from

$$H(q) = -\sum_{t=1}^{T+1} \mathbb{E}_q[\log q(\mathbf{y}_{t-1}|\mathbf{y}_t, \mathbf{x})] = \sum_{t=1}^{T+1} \mathbb{E}_q[\frac{d}{2}(1 + \log(2\pi)) + \frac{1}{2}\log|\Sigma_\phi(\mathbf{y}_t, \mathbf{x})|] \tag{11}$$

where $d$ is the dimension of $\mathbf{y}$ and we use the notation $\mathbf{y}_{T+1} = \mathbf{x}$. The right-hand term can be approximated using Monte Carlo; it is also common to leave the variance $\Sigma_\phi$ fixed (as we typically do in our experiments), in which case $H(q)$ is a constant.

## 4 EXTENSIONS

### 4.1 SEMI-SUPERVISED LEARNING

Following Makhzani et al. (2015), we extend our algorithm to the semi-supervised learning setting where some data points are labeled and we define $p(\mathbf{z}, l) = p(\mathbf{z}|c)p(l)$. Then, the model can be specified as $p_\theta(\mathbf{x}, \mathbf{y}, \mathbf{z}, l) = p_\theta(\mathbf{x}|\mathbf{z}, l)r(\mathbf{y}|\mathbf{z}, l)p_\theta(\mathbf{z}|l)p(l)$ and the variational distributions are $q_\phi(\mathbf{z}|\mathbf{x}, \mathbf{y}, l), q_\phi(\mathbf{y}|\mathbf{x}), q_\phi(l|\mathbf{x})$. In this setting, we consider two cases of whether the label can be observed or not (Kingma et al., 2014). We extend Equation (7) to incorporate the label $l$ corresponding to a data point as follows:

$$\mathcal{L}_{\text{semi}} = \mathbb{E}_{q_\phi(\mathbf{y}, \mathbf{z}|\mathbf{x}, l)}[\log p_\theta(\mathbf{x}|\mathbf{z}, l)] - D_{\text{KL}}(q_\phi(\mathbf{y}, \mathbf{z}|\mathbf{x}, l)||p_\theta(\mathbf{y}, \mathbf{z}|l)) - D_{\text{KL}}(p_\theta(\mathbf{z}|l)||q_\phi(\mathbf{z}|\mathbf{x}, l))$$

When the label $c$ cannot be observed, we treat it as a latent variable and modify the learning objective $\mathcal{U}_{\text{semi}} = \sum_c q_\phi(l|\mathbf{x})\mathcal{L}_{\text{semi}}(\mathbf{x}, l, \theta, \phi) + D_{\text{KL}}(q_\phi(l|\mathbf{x})||p(l))$. Therefore, we can conclude a marginal likelihood on our dataset as follows: $\tilde{\mathcal{L}}_{\text{semi}} = \sum_{(\mathbf{x}, l) \in L} \mathcal{L}_{\text{semi}}(\mathbf{x}, l, \theta, \phi) + \sum_{\mathbf{x} \in U} \mathcal{U}_{\text{semi}}(\mathbf{x}, \theta, \phi)$. where $L$ and $U$ are the sets of data with and without labels, respectively.

We also want to guarantee that all model parameters can be learned in all cases, including $q_\phi(l|\mathbf{x})$, such that this posterior can be applied as a classifier during inference. Thus, we combine the marginal likelihood with a classification loss to form an extended learning objective: $\tilde{\mathcal{L}}_{\text{semi}_\alpha} = \tilde{\mathcal{L}}_{\text{semi}} + \boldsymbol{\alpha} \cdot \mathbb{E}_{\tilde{p}(\mathbf{x}, l)}[-\log q_\phi(l|\mathbf{x})]$

### 4.2 CLUSTERING

We have further extended our algorithm to encompass the clustering paradigm. We propose two distinct strategies. In the first approach, we simply set $p_\theta(\mathbf{z})$ as a mixture of desired priors. The

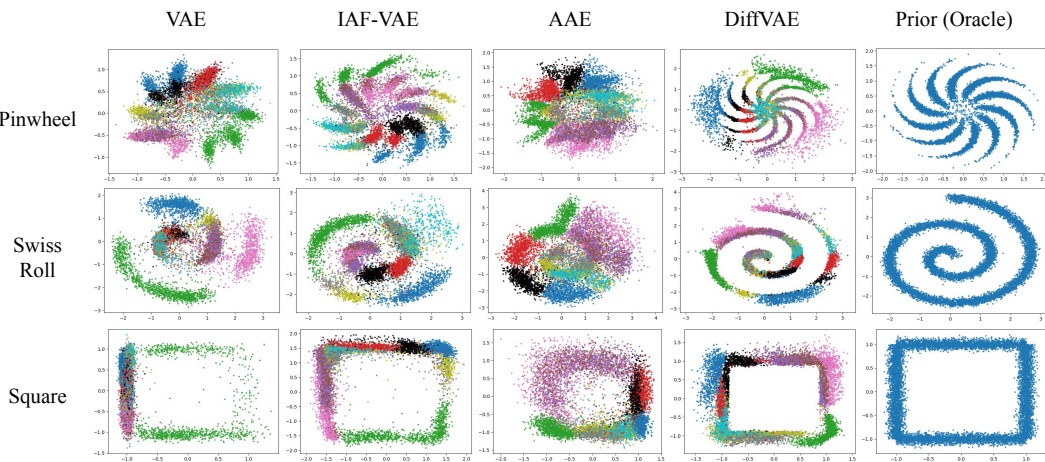

Figure 2: Unsupervised visualization on MNIST using three different priors (pinwhell, swiss roll, and square). Each color indicates a class.

means of these priors are characterized by $\theta$. From these means, cluster membership, denoted as $\mathbf{c}$ can be deduced. Remarkably, this approach requires no alteration to the existing learning objective.

Alternatively, the second method retains the original prior but introduces an additional cluster latent variable $\mathbf{c}$ where $\sum_i c_i = 1$. Thus, the model can be specified as $p_{\theta}(\mathbf{x}, \mathbf{y}, \mathbf{z}, \mathbf{c}) = p_{\theta}(\mathbf{x}|\mathbf{z}, \mathbf{c})r(\mathbf{y}|\mathbf{z})p_{\theta}(\mathbf{z})p(\mathbf{c})$ with $p(\mathbf{c}) = Dir(\epsilon)$. Consequently, the variational distributions become $q_{\phi}(\mathbf{z}|\mathbf{y}, \mathbf{c}, \mathbf{x}), q_{\phi}(\mathbf{y}, \mathbf{c}|\mathbf{x})$. This reformulates the learning objective as:

$$\mathcal{L}_{\text{clus}}(\mathbf{x}) = \mathbb{E}_{q_{\phi}(\mathbf{y}, \mathbf{z}, \mathbf{c}|\mathbf{x})}[\log p_{\theta}(\mathbf{x}|\mathbf{z}, \mathbf{c})] - D_{\text{KL}}(q_{\phi}(\mathbf{y}, \mathbf{z}, \mathbf{c}|\mathbf{x})||p_{\theta}(\mathbf{y}, \mathbf{z}, \mathbf{c})) - D_{\text{KL}}(p_{\theta}(\mathbf{z})||q_{\phi}(\mathbf{z}|\mathbf{x}))$$

## 5 EXPERIMENTS

We compare DiffVAE with vanilla VAE (Kingma & Welling, 2013), IAF-VAE (Kingma et al., 2016), and AAE (Makhzani et al., 2015) on MNIST (Lecun et al., 1998) and CIFAR-10 (Krizhevsky & Hinton, 2009) in unsupervised and semi-supervised learning settings, and also on the 1000 genomes dataset Siva (2008) in clustering settings. The priors, model architecture, and training details can be founded in Appendix A.3, Appendix A.4, and Appendix A.5 respectively. All results are reported with 95% confidence interval using 5 different seeds.

### 5.1 UNSUPERVISED LEARNING

We fit a model $p_{\theta}(\mathbf{x}, \mathbf{z})$ on the MNIST and CIFAR-10 datasets with three priors $p(\mathbf{z})$: pinwheel, swiss roll, and square and report our results in Table 1 and Table 4. We measure a k-nearest neighbors classification accuracy of the latents (Acc). We also measure latent negative log-likelihood (Latent NLL) by fitting a kernel density estimator (KDE) on the latents produced by the model with test data as input and compute the log-likelihood of the latents sampled from the prior under the fitted KDE.

We consistently see our method perform best on both KNN accuracy and latent NLL, indicating that its output latents are the most informative of labels and best-aligned with the priors. Figure 2 reports qualiatitve results that indicate that DiffVAE enforces most accurately the correct prior shape. Figure 3 shows the decoded images of each method trained with a square prior. The input latents are sampled by walking along the square. DiffVAE generates a more diverse collection of digits (all classes) with smoother transitions.

### 5.2 SEMI-SUPERVISED LEARNING

We also evaluate the performance of our method and the baselines under semi-supervised learning setting where some labels are observed (1,000 for MNIST and 10,000 for CIFAR-10) and the partitions of the priors are known. We use the same set of priors and baselines. Details on how we partition each prior into $p_{\theta}(\mathbf{z}|l)$ can be founded in Appendix A.3. The partitions defined for our

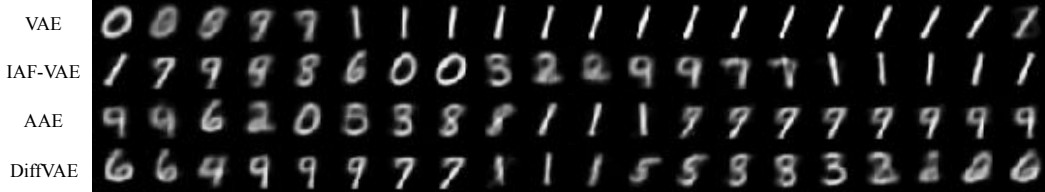

Figure 3: Reconstructed images from models trained with square prior. The input latents are sampled by walking along the square.

| Method | Pinwheel | | Swiss Roll | | Square | |
|---|---|---|---|---|---|---|
| | Acc | Latent NLL | Acc | Latent NLL | Acc | Latent NLL |
| VAE | $0.58 \pm 0.01$ | $6.18 \pm 0.93$ | $0.56 \pm 0.055$ | $82.96 \pm 99.30$ | $0.35 \pm 0.02$ | $25.07 \pm 62.41$ |
| IAF-VAE | $\mathbf{0.71 \pm 0.00}$ | $2.06 \pm 0.21$ | $\mathbf{0.67 \pm 0.02}$ | $12.13 \pm 18.28$ | $0.65 \pm 0.03$ | $1.53 \pm 0.82$ |
| AAE | $0.67 \pm 0.02$ | $1.94 \pm 0.09$ | $\mathbf{0.68 \pm 0.03}$ | $3.43 \pm 0.14$ | $0.57 \pm 0.02$ | $1.97 \pm 0.70$ |
| Ours | $\mathbf{0.72 \pm 0.01}$ | $\mathbf{1.57 \pm 0.10}$ | $0.66 \pm 0.01$ | $3.19 \pm 0.07$ | $\mathbf{0.72 \pm 0.03}$ | $\mathbf{1.24 \pm 0.13}$ |

Table 1: Unsupervised learning on MNIST. We report accuracy using KNN (K=20) classifier and latent negative log-likelihood with pinwheel, swiss roll, and square priors.

priors are local parts of the priors. We substitute the KNN accuracy metric with the average class latent NLL, which measures how well the test latents from each class match their respective partition of the prior $p_{\boldsymbol{\theta}}(\mathbf{z}|l)$. In our case, this metric implicitly measures accuracy because the metric requires that the test latents for each class be close to each other, as the partitions defined are local parts of the full prior. The results are shown in Table 2 and Table 5. DiffVAE mostly outperforms the baselines across different priors and metrics. We also show the visualization in Figure 4 where DiffVAE matches the prior almost perfectly.

| Method | Pinwheel | | Swiss Roll | | Square | |
|---|---|---|---|---|---|---|
| | Latent NLL | Avg Class Latent NLL | Latent NLL | Avg Class Latent NLL | Latent NLL | Avg Class Latent NLL |
| VAE | $7.77 \pm 1.83$ | $15.64 \pm 8.79$ | $56.91 \pm 33.90$ | $244.51 \pm 94.19$ | $1.30 \pm 0.03$ | $-0.71 \pm 0.05$ |
| IAF-VAE | $2.71 \pm 0.22$ | $1.84 \pm 0.46$ | $8.10 \pm 12.17$ | $14.59 \pm 31.88$ | $1.04 \pm 0.04$ | $-1.01 \pm 0.03$ |
| AAE | $1.90 \pm 0.04$ | $0.64 \pm 0.11$ | $\mathbf{3.35 \pm 0.17}$ | $1.86 \pm 0.77$ | $1.70 \pm 0.26$ | $2.50 \pm 4.70$ |
| DiffVAE | $\mathbf{1.28 \pm 0.01}$ | $\mathbf{-0.94 \pm 0.01}$ | $\mathbf{3.35 \pm 0.10}$ | $\mathbf{1.24 \pm 0.09}$ | $\mathbf{0.96 \pm 0.03}$ | $\mathbf{-1.20 \pm 0.02}$ |

Table 2: Semi-supervised learning on MNIST (1,000 labels). We report average class latent negative log-likelihood and latent negative log-likelihood with pinwheel, swiss roll, and square priors.

## 5.3 CLUSTERING AND VISUALIZATION FOR GENOTYPE ANALYSIS

In this section, we report results on an real-world task in genome analysis. Visualizing genotype data reveals patterns in the latent ancestry of individuals. We compare DiffVAE against with the three strong clustering baselines using the 1000 Genomes dataset. We also report visualizations from three dimensionality reduction algorithms: PCA, TSNE, and UMAP. For each clustering algorithm, we seek to discover up to 20 clusters. We report quantitative results in terms of cluster purity, cluster completeness, and normalized mutual information (NMI). There is an inherent trade-off between cluster purity completeness. The overall clustering performance can be captured with NMI.

In Table 3, we see that DiffVAE attains the best performance on cluster purity and NMI. For cluster completeness, VAE and AAE have better means but much larger confidence interval. Furthermore, we visualize our genotype clustering results in latent space, shown in Figure 5, and also report results from classical di-

| Method | Cluster Purity | Cluster Completeness | NMI |
|---|---|---|---|
| VAE | $0.28 \pm 0.02$ | $\mathbf{0.78 \pm 0.16}$ | $0.59 \pm 0.08$ |
| IAF-VAE | $0.29 \pm 0.04$ | $0.73 \pm 0.06$ | $0.55 \pm 0.06$ |
| AAE | $0.37 \pm 0.06$ | $0.76 \pm 0.11$ | $0.63 \pm 0.02$ |
| DiffVAE | $\mathbf{0.45 \pm 0.03}$ | $0.75 \pm 0.05$ | $\mathbf{0.66 \pm 0.04}$ |

Table 3: Quantitative genotype clustering results.

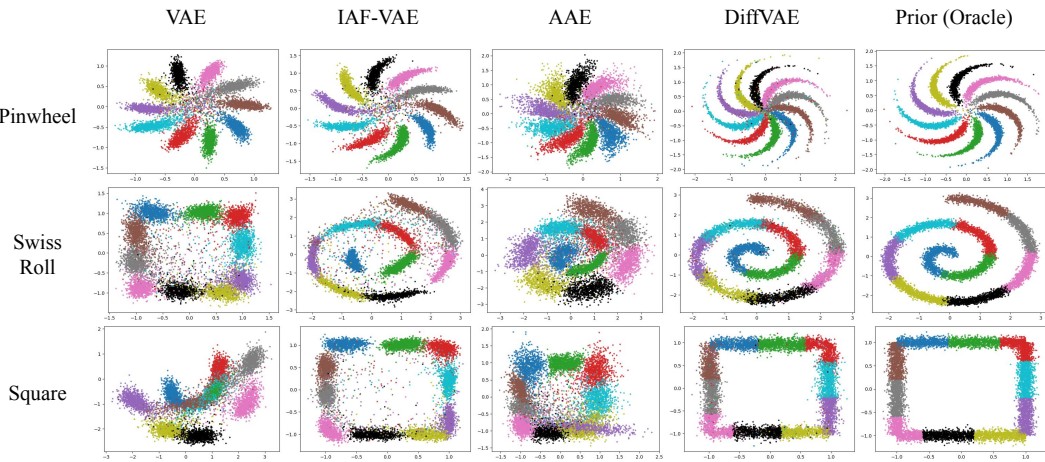

Figure 4: Semi-supervised visualization on MNIST with 1,000 labels using three different priors (pinwheel, swiss roll, and square). Each a indicates one class.

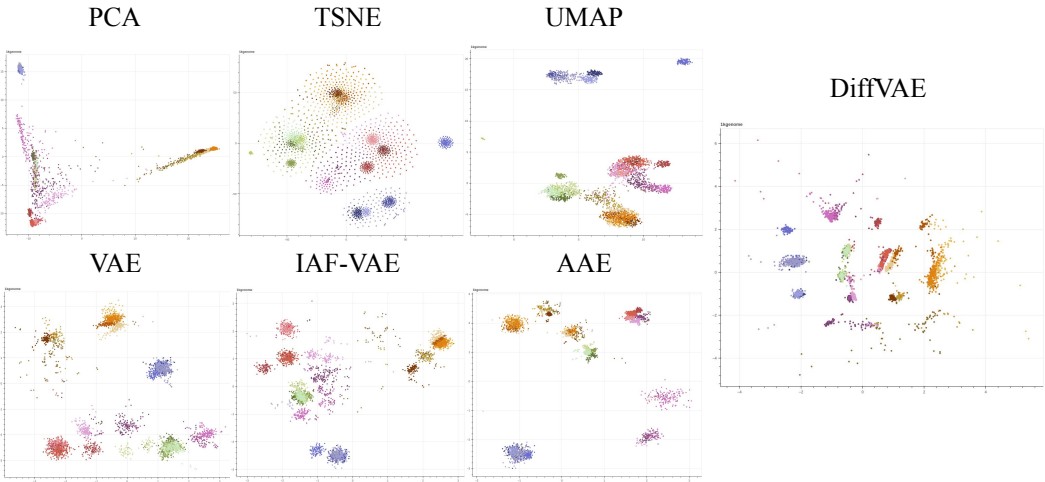

Figure 5: Visualization of genotype clusters. A color represents one ethnicity.

mensionality reduction and visualization methods that do not perform clustering (PCA (Wold et al., 1987), t-SNE (Van der Maaten & Hinton, 2008), and UMAP (McInnes et al., 2018)). The legend of Figure 5 can be founded at Figure 6.

# 6 RELATED WORK, DISCUSSION, AND CONCLUSION

There are a number of recent works merging VAE and diffusion models. Vahdat et al. (2021); Wehenkel & Louppe (2021) use diffusion with VAE priors. Recent research (Preechakul et al., 2022; Zhang et al., 2022; Wang et al., 2023b) has also melded auto-encoders with diffusion models, focusing on semantically meaningful latents. Cohen et al. (2022) crafts a diffusion bridge linking a continuous coded vector to a non-informative prior distribution. These works focus on sample quality and directly fit $q_\phi(\mathbf{z}|\mathbf{x})$ to a complex distribution (e.g., compressed images (Rombach et al., 2022)); our work instead seeks to achieve high sample quality in latent space, and obtains complex $\mathbf{z}$ by diffusion from a simple $\mathbf{y} \sim q_\phi(\mathbf{y}|\mathbf{x})$.

Latent variable models in general are an attractive alternative to visualization methods like PCA, UMAP, and t-SNE (McInnes et al., 2018; Van der Maaten & Hinton, 2008). Domain-specific knowledge can be injected through the prior, and deep neural networks can be utilized to achieve a more expressive mapping from the data space to the latent space. Nevertheless, downsides of LVMs are that they are more computationally expensive and require careful hyperparameter tuning.

While this paper focuses on applications of DiffVAE to dimensionality reduction and visualization, there exist other tasks for the algorithm, e.g., density estimation or sample quality. Since our learning objective differs from the ELBO (it adds a regularizer), we anticipate gains on models whose training benefits from regularization, but perhaps not on all models. Also, attaining competitive likelihood estimation requires architecture improvements that are orthogonal to this paper. However, our ability to generate diverse samples and achieve class separation in latent space hints at the method's potential on these tasks. Improve variational inference also holds promise to improve downstream applications of generative modeling, e.g., decision making or causal effect estimation (Nguyen & Grover, 2022; Deshpande et al., 2022; Deshpande & Kuleshov, 2023; Rastogi et al., 2023).

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

# A APPENDIX

## A.1 AUXILIARY-VARIABLE DIFFUSION ENCODERS

Denoising variational inference can also be understood from the perspective of auxiliary-variable generative models (Maaløe et al., 2016). Our method can be seen as introducing an approximate posterior $q_{\phi}(\mathbf{z}|\mathbf{x}) = \int_{\mathbf{y}} q_{\phi}(\mathbf{z}, \mathbf{y}|\mathbf{x})$ which itself contains latent variables $\mathbf{y}$.

The standard approach to fit auxiliary-variable generative models is to apply the ELBO twice:

$$\log p_{\boldsymbol{\theta}}(\mathbf{x}) \geq \log p_{\boldsymbol{\theta}}(\mathbf{x}) - D_{\mathrm{KL}}(q_{\phi}(\mathbf{z}|\mathbf{x})||p_{\boldsymbol{\theta}}(\mathbf{z}|\mathbf{x})) = \mathrm{ELBO}_{\mathbf{z}|\mathbf{x}}(\mathbf{x}, \boldsymbol{\theta}, \phi) \quad (12)$$

$$\geq \mathrm{ELBO}_{\mathbf{z}|\mathbf{x}}(\mathbf{x}, \boldsymbol{\theta}, \phi) - \mathbb{E}_{q_{\phi}(\mathbf{z}|\mathbf{x})}[D_{\mathrm{KL}}(q_{\phi}(\mathbf{y}|\mathbf{x}, \mathbf{z})||r(\mathbf{y}|\mathbf{x}, \mathbf{z}))] \quad (13)$$

$$= \mathbb{E}_{q_{\phi}(\mathbf{y}, \mathbf{z}|\mathbf{x})}[\log p_{\boldsymbol{\theta}}(\mathbf{x}, \mathbf{z})r(\mathbf{y}|\mathbf{x}, \mathbf{z}) - \log q(\mathbf{y}, \mathbf{z}|\mathbf{x})], \quad (14)$$

where in Equation (13) we applied the ELBO over $\mathbf{z}$, and in Equation (12) we applied the ELBO again over the latent $\mathbf{y}$ of $q$. The $r(\mathbf{y}|\mathbf{x}, \mathbf{z})$ can be interepreted as *an approximate variational posterior* for the true posterior $q_{\phi}(\mathbf{y}|\mathbf{x}, \mathbf{z})$ of the data distribution. Specifically, the gap between Equation (12) and $\log p_{\boldsymbol{\theta}}(\mathbf{x})$ is precisely $D_{\mathrm{KL}}(q_{\phi}(\mathbf{z}|\mathbf{x})||p_{\boldsymbol{\theta}}(\mathbf{z}|\mathbf{x})) + \mathbb{E}_{q_{\phi}(\mathbf{z}|\mathbf{x})}[D_{\mathrm{KL}}(q_{\phi}(\mathbf{y}|\mathbf{x}, \mathbf{z})||r(\mathbf{y}|\mathbf{x}, \mathbf{z}))]$. Thus, if we correctly match the pairs of approximate posteriors, we will achieve a tight bound.

Additionally, notice that Equation (14) is exactly the ELBO in Equation (2) after rearranging terms. Thus the two perspectives are equivalent: using variational inference to fit an augmented model $p_{\boldsymbol{\theta}}(\mathbf{x}, \mathbf{y}, \mathbf{z})$ is equivalent to fitting the original $p_{\boldsymbol{\theta}}(\mathbf{x}, \mathbf{z})$ and applying the ELBO to $p_{\boldsymbol{\theta}}$, followed by applying the ELBO again to the latent $\mathbf{y}$ in $q$. Since our objective $\mathcal{L}$ is Equation (14)) augmented with $\mathcal{L}_{\mathrm{sleep}}$, denoising VI can also be interpreted as fitting an latent-variable approximate posterior $q$. As seen in Equation (9), we implicitly minimize the reverse KL divergence $\mathbb{E}_{p_{\boldsymbol{\theta}}(\mathbf{z})}[D_{\mathrm{KL}}(r(\mathbf{y}|\mathbf{z})||q_{\phi}(\mathbf{y}|\mathbf{z}, \mathbf{x}))]$.

Note, however, that there is a crucial difference between our method and auxiliary-variable generative models: we do not optimize the parameters of $r$. Rather, $r$ is a user-defined noising process that acts as a regularizer for the model $q_{\phi}$ and whose reverse we seek to fit.

## A.2 Regularizing Variational Inference with Denoising Processes

Lastly, our approach can be understood from a third perspective: that of regularization. As we mentioned earlier, $\mathcal{L}$ is the ELBO with an additional regularization term that encourages the approximate posterior $q(\mathbf{z}|\mathbf{x}, \mathbf{y})q(\mathbf{y}|\mathbf{x})$ to gradually transform $\mathbf{y}$ into $\mathbf{z}$ following the guidance of $r$. In that sense, $r$ provides data augmentation to ensure that the mapping from $\mathbf{y}$ to $\mathbf{x}$ is easier to learn, similarly to how a diffusion process provides additional learning signal for generating $\mathbf{x}$ from noise.

Our framework also supports the user defining general distributions as regularizers $r_1(\mathbf{z}, \mathbf{y})$ and $r_2(\mathbf{y}|\mathbf{z}, \mathbf{x})$. Here, $r_1$ puts a custom prior on $\mathbf{z}, \mathbf{y}$, while $r_2$ specifies a mapping between $\mathbf{y}, \mathbf{z}$ that should be followed by the encoder model. We write $\mathcal{L}_{\text{reg}}(\mathbf{x}, \boldsymbol{\theta}, \boldsymbol{\phi}, r_1)$ and $\mathcal{L}_{\text{sleep}}(\mathbf{x}, \boldsymbol{\phi}, r_2)$, where each term is still defined as in Equation (7) and Equation (6), except using $r_1, r_2$ and weoptimize

$$\mathcal{L}(\mathbf{x}, \boldsymbol{\theta}, \boldsymbol{\phi}, r_1, r_2) = \mathcal{L}_{\text{rec}}(\mathbf{x}, \boldsymbol{\theta}, \boldsymbol{\phi}) + \beta \cdot \mathcal{L}_{\text{reg}}(\mathbf{x}, \boldsymbol{\theta}, \boldsymbol{\phi}, r_1) + \gamma \cdot \mathcal{L}_{\text{sleep}}(\mathbf{x}, \boldsymbol{\phi}, r_2), \qquad (15)$$

where $\beta, \gamma > 0$ (in particular, $\beta$ mirrors the $\beta$-VAE model (Higgins et al., 2016)).

Examples of $r_2$ include Gaussian diffusion, discrete noising processes, as well as noise conditioned on $\mathbf{x}$. Note that $r_2$ could even be *fully deterministic* when parameterized with a bijective flow. With a strong regularizer, we could set $q(\mathbf{z}|\mathbf{y}, \mathbf{x})$ to equal the reverse $r_2(\mathbf{z}|\mathbf{y}, \mathbf{x})$ of this flow. Thus, for suitably designed $r_2$, we could perfectly match the shape of any mapping. This mirrors the InteL-VAE framework (Miao et al., 2021), with support for flexible and probabilistic regularizers.

Most importantly, our method can be understood as introducing a new form of regularization. While a standard VAE introduces a prior regularization term, we introduce a new form of regularization via a function $r_2$. This can be advantageous when specifying or enforcing certain priors is difficult.

## A.3 Priors

Below we describe the sampling process for each prior.

**Pinwheel.** This distribution was used in (Johnson et al., 2016). We define the number of clusters to be 10. For semi-supervised learning experiments, this prior is partitioned into 10 partitions, each partition being a cluster.

**Swiss Roll.** This distribution was used in Marsland (2014). We add noise $\sigma = 0.1$ to the prior. For semi-supervised learning experiments, this prior is partitioned into 10 partitions. The samples from the prior, before we add noise to it, can actually be characterized by a single scalar representing how far you are long the swiss roll from the center. The paritioning is done by creating 10 equal-length intervals in this 1D space.

**Square.** This distribution has the shaped of a square going from -1 to 1 in both axes. Each position on the square can be characterized by a single scalar representing how far you are from the top left corner. Sampling is done by sampling the position uniformly and turn the 1D position to 2D latent. We add noise $\sigma = 0.06$ to the prior. For semi-supervised learning experiments, this prior is partitioned into 10 partitions. The partitioning is done by creating 10 equal-length intervals in the 1D position space.

VAE and IAF-VAE requires that we can evaluate the prior density. To do this, for all priors, we evaluate the density by fitting a kernel density estimator with mixture of gaussian kernel with bandwidth equal to 0.005, 0.008, 0.01, 0.03, and 0.05.

## A.4 Model Architecture

All methods use the same architecture for encoder $q_{\boldsymbol{\phi}}(z|x)$ and decoder $p_{\boldsymbol{\theta}}(x|z)$, excluding the extra parts specific to each method which we describe below, for the same dataset. For MNIST, the encoder and decoder are multi-layer perceptron with two hidden layers, each with 1000 hidden units. For CIFAR-10, the encoder is a 4-layer convolutional neural network with (16, 32, 64, 128) channels with a linear layer on top, and the decoder is a 4-layer tranposed convolutional neural network with (64, 32, 16, 3) channels where a linear layer is used to first turn the feature dimension from 2 to 64.

**IAF-VAE** employs 4 IAF transformations on top of the encoder, each is implemented with a 4-layer MADE. The number of hidden units in MADE is 24. The ordering is reversed between every other IAF transformation.

Figure 6: Legend showing what ethnicity each color corresponds to in the 1000 Genomes dataset

**AAE** has a discriminator, used in adversarial training, which is a multi-layer perceptron with two hidden layers, each with 1000 hidden units.

**DiffVAE** has a diffusion model on top of the encoder. The time-conditioned reverse diffusion distribution is implemented with a 5-layer time-conditioned multi-layer perceptron, each with 128 hidden units. A time-conditioned linear layer learns an additional embedding for each timestep and adds it to the output of the linear layer.

## A.5  TRAINING DETAILS

For training, we use Adam optimizer and latent size of 2 for all of our experiments. The training details of each algorithm are detailed below:

**VAE.** The batch size is set to 128. The number of epochs is 200 for unsupervised and clustering experiments and 50 for semi-supervised experiments. The learning rate is 0.0001. The loss is BCE for MNIST and CIFAR-10 experiments and MSE for genotype analysis experiments. For semi-supervised MNIST experiments, the kl divergence weight is set to be 0.01, while for semi-supervised CIFAR-10 experiments, the kl divergence weight is set to be 0.01. For other experiments, the KL divergence weight is set with a schedule linear on number of epochs going from 0 to 0.01. We also have a weight of 5 multiplied to the prior density.

**IAF-VAE.** The batch size, number of epochs, learning rate, loss, KL divergence weight, and prior density weight are the same as VAE. The context size, i.e., the size of features used to initialize the flow layers for different datat point, is 10.

**AAE.** The batch size is set to 128. The number of epochs is 200 for all experiments. The learning rate is 0.0002. The loss is MSE for all experiments. To stabilize the training, we add noise to the input to the discriminator with sigma 0.3 at the start and lower it by 0.1 for every 50 epochs. The noise equals to 0 at epoch 150.

**DiffVAE.** The batch size is set to 128 for most experiments, except for semi-supervied experiments where the batch size is 1024. The number of epochs is 200 for unsupervised and clustering experiments and 30 for semi-supervised experiments. The learning rate is 0.0001. The loss is BCE for MNIST and CIFAR-10 experiments and MSE for genotype analysis experiments. For unsupervised MNIST and CIFAR-10 experiments, the KL divergence weight is set to 0.003. For semi-supervised MNIST experiment, we use KL divergence weight of 0.1. For semi-supervised CIFAR-10 experiment, we use KL divergence weight of 0.5. For clustering experiment, we use KL divergence weight of 0.005. The number of timesteps is 20 for unsupervised and clustering experiments and 100 for semi-supervised experiments.

## A.6  GENOTYPE ANALYSIS EXPERIMENTS DETAILS

Before inputting the data points into any of the visualization methods, we first pre-process it by running a PCA and keep only the first 1000 principal components of the data points. We further divide the features by 30 for all latent variables model methods.

The legend of the 1000 Genomes Visualization plot can be found at Figure 6.

| Method | Pinwheel | | Swiss Roll | | Square | |
|---|---|---|---|---|---|---|
| | Acc | Latent NLL | Acc | Latent NLL | Acc | Latent NLL |
| VAE | $0.16 \pm 0.01$ | $86.9 \pm 99.03$ | $0.15 \pm 0.01$ | $360.41 \pm 185.87$ | $0.15 \pm 0.01$ | $63.3 \pm 70.64$ |
| IAF-VAE | $0.19 \pm 0.01$ | $3.12 \pm 1.16$ | $0.16 \pm 0.01$ | $189.39 \pm 57.48$ | $0.18 \pm 0.01$ | $0.99 \pm 0.11$ |
| AAE | $0.23 \pm 0.00$ | $2.30 \pm 0.07$ | $\mathbf{0.23 \pm 0.01}$ | $\mathbf{2.84 \pm 0.08}$ | $0.20 \pm 0.01$ | $1.46 \pm 0.49$ |
| DiffVAE | $\mathbf{0.23 \pm 0.01}$ | $\mathbf{1.37 \pm 0.02}$ | $\mathbf{0.23 \pm 0.01}$ | $\mathbf{2.80 \pm 0.11}$ | $\mathbf{0.24 \pm 0.01}$ | $\mathbf{0.86 \pm 0.05}$ |

Table 4: Unsupervised learning on CIFAR-10. We report accuracy using KNN (K=20) classifier and latent negative log-likelihood with pinwheel, swiss roll, and square priors.

| Method | Pinwheel | | Swiss Roll | | Square | |
|---|---|---|---|---|---|---|
| | Latent NLL | Avg Class Latent NLL | Latent NLL | Avg Class Latent NLL | Latent NLL | Avg Class Latent NLL |
| VAE | $7.67 \pm 1.34$ | $9.84 \pm 2.87$ | $49.64 \pm 28.12$ | $128.46 \pm 47.08$ | $1.51 \pm 0.23$ | $0.65 \pm 0.37$ |
| IAF-VAE | $4.54 \pm 0.59$ | $4.46 \pm 0.99$ | $64.27 \pm 73.87$ | $102.16 \pm 106.36$ | $1.04 \pm 0.07$ | $0.07 \pm 0.09$ |
| AAE | $2.33 \pm 0.02$ | $2.42 \pm 0.06$ | $3.76 \pm 0.04$ | $4.23 \pm 0.13$ | $1.45 \pm 0.12$ | $1.31 \pm 0.20$ |
| DiffVAE | $\mathbf{1.49 \pm 0.09}$ | $\mathbf{0.35 \pm 0.18}$ | $\mathbf{3.35 \pm 0.27}$ | $\mathbf{2.28 \pm 0.39}$ | $\mathbf{0.97 \pm 0.03}$ | $\mathbf{-0.29 \pm 0.04}$ |

Table 5: Semi-supervised learning on CIFAR-10 (10,000 labels). We report average class latent negative log-likelihood and latent negative log-likelihood with pinwheel, swiss roll, and square priors.

