# OpenReview forum: "Denoising Diffusion Variational Inference"
_ICLR.cc/2024/Conference — Submitted to ICLR 2024_

### Official Review · Reviewer_3sxG · 2023-10-30

**Soundness:** 2 fair
**Presentation:** 2 fair
**Contribution:** 2 fair
**Rating:** 3
**Confidence:** 3

**Summary:**

This paper proposes denoising variational inference. Specifically, they introduce a  user-specified noising process, which transfers a latent variable $z$ from an intractable posterior to another easy to model variable $y$. Then, the authors propose a lower bound on the marginal likelihood inspired by wake-sleep. The method is then extended to the diffusion-based encoders, which can also be viewed as adding a new form of regularization. The authors also discuss some extension topics including semi-supervised learning and clustering.

**Strengths:**

This work propose a novel variational lower bound on the marginal likelihood. Besides, the analysis and experiments in dimension reduction are interesting and meaningful.

**Weaknesses:**

1. I see the point of adding auxiliary variables to the VAE encoder, but I can't find a strong reason to introduce auxiliary variables in diffusion models.
2. Both Sec. 3.1 and 3.2 are introducing the main method. But it is hard to tell what I should take away for each section.
3. The experiments are comparing with classical VAE-based methods. It would be more convincing if more advanced VAE-based methods are considered.
4. In Fig. 3 of the experiments, using a more complicated prior does not improve the  visual quaility, and even don't see more diversity compared with AAE.

**Questions:**

1. Can you point out the difference between DiffVAE and Variational diffusionn models?
2. Some other methods like two stage VAE[1] are also learning a more complicated prior (without knowing the true density), then what is the strength of DiffVAE over these methods?
3. Can you provide rigorous convergence and error analysis for your proposed bound?
3. What is the time consumption for the experiments? Compared with classical models?

[1] Dai B, Wipf D. Diagnosing and enhancing VAE models[J]. arXiv preprint arXiv:1903.05789, 2019.

---

> ### Author Response · Authors · 2023-11-21
> **Response to Reviewer 3sxG (Part 1)**
>
> We want to thank the reviewer for the thoughtful review and constructive feedback. We address each of your comments below:
>
> ***
> > **Q1** I can't find a strong reason to introduce auxiliary variables in diffusion models
>
> **A1** The motivation of introducing auxiliary variables is to increase the flexibility of the variational distribution that can thereby improve the variational lower bound. This has been an important research direction for variational inference and many previous works have demonstrated its usefulness [1, 2, 3, 4].
>
> To clarify, we are not introducing auxiliary variables in the sense of augmenting diffusion models; we introduce them in the approximate posterior as in VAE with auxiliary variables (equivalently, hierarchical VAE (HVAE)), where the relationship between the auxiliary latent $\mathbf{y}$ and the main latent $\mathbf{z}$ is defined as diffusion process.
>
> We add a new table here that compares different baselines, including a new HVAE baseline, with our method
>
> **Comparison of DiffVAE to other relevant methods**
> | Model   | Training Objective | Approximating Family | Sample-based Prior | Auxiliary Variable | Tasks | Simplified Graphical Illustration |
> |---|---|---|---|---|---|---|
> | VAE 	| ELBO           	| Diagonal Gaussian	| No             	| No             	| Density estimation | $\mathbf{x}$ -> $\mathbf{z}$ -> $\mathbf{x}$ |
> | IAF-VAE | ELBO           	| Normalizing flow 	| No             	| Yes            	| Density estimation/Visualization | $\mathbf{x}$ -> $\mathbf{z}_0$ -> $\mathbf{z}_T$ -> $\mathbf{x}$ |
> | AAE 	| Adversarial training | Adversarial generator | Yes           	| No             	| Visualization | $\mathbf{x}$ -> $\mathbf{z}$ -> $\mathbf{x}$ |
> | HVAE w/o and w/ flow | ELBO | Factorial Normal / Normalizing flow | No 	| Yes           	| Density estimation/High-quality sample generation | $\mathbf{x}$ -> $\mathbf{z}_0$ -> $\mathbf{z}_T$ -> $\mathbf{z}_0$ -> $\mathbf{x}$ |
> | ADGM	| ELBO           	| Non-Gaussian     	| No             	| Yes            	| Density estimation | $\mathbf{x}$ -> $\mathbf{a}$ -> $\mathbf{z}$ -> $\mathbf{x}$ |
> | LDM 	| ELBO           	| Diagonal Gaussian	| No             	| Yes            	| High-quality sample generation | $\mathbf{x}$ -> $\mathbf{z}_0$ -> $\mathbf{z}_T$ -> $\mathbf{z}_0$ -> $\mathbf{x}$ |
> | LSGM	| ELBO+score matching | Diagonal Gaussian   | No             	| Yes            	| High-quality sample generation | $\mathbf{x}$ -> $\mathbf{z}_0$ -> $\mathbf{z}_T$ -> $\mathbf{z}_0$ -> $\mathbf{x}$ |
> | DiffVAE | ELBO+sleep term	| Denoising diffusion | Yes            	| Yes            	| Density estimation/Visualization | $\mathbf{x}$ -> $\mathbf{z}_T (\mathbf{y})$ -> $\mathbf{z}_0 (\mathbf{z})$ -> $\mathbf{x}$ |
>
>
> [1] Rezende, Danilo, and Shakir Mohamed. "Variational inference with normalizing flows." International conference on machine learning. PMLR, 2015.
>
> [2] Barber, David, and Felix Agakov. "The im algorithm: a variational approach to information maximization." Advances in neural information processing systems 16.320 (2004): 201.
>
> [3] Maaløe, Lars, et al. "Auxiliary deep generative models." International conference on machine learning. PMLR, 2016.
>
> [4] Sønderby, Casper Kaae, et al. "Ladder variational autoencoders." Advances in neural information processing systems 29 (2016).
>
> ***
> > **Q2** Both Sec. 3.1 and 3.2 are introducing the main method. But it is hard to tell what I should take away for each section.
>
> **A2** We present two perspectives to understanding our denoising diffusion variational inference in Sections 3.1 and 3.2. We recognize that having section 3.2 (and 3.3) in the main text might affect the readability of the paper and have moved it to the appendix so that it is an optional read. More details on both sections can be found below:
>
> Section 3.1 introduces DiffVAE, which fits a latent variable model with a complex prior by extending the latent space with auxiliary latents through a noising process $r(\mathbf{y} | \mathbf{z})$. This process simplifies modeling the posterior of $\mathbf{y}$. To overcome the weak learning signals from the standard ELBO, we propose an augmented ELBO with an additional regularizer, $\mathcal{L}_{sleep} (\phi)$. This involves a diffusion-like training procedure, optimizing the reverse denoising process and leveraging the wake-sleep algorithm for efficient training in latent space.
>
> In Section 3.2, we draw a parallel between DiffVAE and auxiliary variable VAEs, including hierarchical VAEs with normalizing flows. Here, we use an approximate posterior $q_\phi (\mathbf{z} | \mathbf{x})$ with latent variables $\mathbf{y}$. This setup equates fitting an augmented model $p_\theta (\mathbf{x}, \mathbf{y}, \mathbf{z})$ with the original model $p_\theta (\mathbf{x}, \mathbf{z})$ using a double application of the ELBO. This approach seeks a tight bound by aligning approximate posteriors and implicitly minimizes the reverse KL divergence in the denoising variational inference process.

---

> ### Author Response · Authors · 2023-11-21
> **Response to Reviewer 3sxG (Part 2)**
>
> ***
> > **Q3** It would be more convincing if more advanced VAE-based methods are considered.
>
> **A3** We add a new HVAE [1, 2] with normalizing flow baseline for unsupervised learning experiments to further highlight the connection between DiffVAE and auxiliary-variable VAE / hierarchical VAE (HVAE). The results on MNIST and CIFAR-10 below show that DiffVAE outperforms the baselines in almost all scenarios:
>
> **Unsupervised learning on MNIST**
> | Method        | Acc (Pinwheel)      | Latent NLL (Pinwheel) | Acc (Swiss Roll)     | Latent NLL (Swiss Roll) | Acc (Square)        | Latent NLL (Square) |
> |---------------|---------------------|-----------------------|----------------------|-------------------------|---------------------|---------------------|
> | VAE           | 0.58 ± 0.01         | 6.18 ± 0.93           | 0.56 ± 0.055         | 82.96 ± 99.30           | 0.35 ± 0.02         | 25.07 ± 62.41       |
> | IAF-VAE       | **0.71 ± 0.00**     | 2.06 ± 0.21           | 0.67 ± 0.02          | 12.13 ± 18.28           | 0.65 ± 0.03         | 1.53 ± 0.82         |
> | AAE           | 0.67 ± 0.02         | 1.94 ± 0.09           | 0.68 ± 0.03          | 3.43 ± 0.14             | 0.57 ± 0.02         | 1.97 ± 0.70         |
> | HVAE w/ flow  | **0.72 ± 0.01**     | 2.18 ± 0.13           | **0.71 ± 0.02**      | 4.17 ± 1.23             | **0.70 ± 0.01**     | **1.25 ± 0.08**     |
> | DiffVAE       | **0.72 ± 0.01**     | **1.57 ± 0.10**       | 0.66 ± 0.01          | **3.19 ± 0.07**         | **0.72 ± 0.03**     | **1.24 ± 0.13**     |
>
> **Unsupervised learning on CIFAR-10**
> | Method        | Acc (Pinwheel)      | Latent NLL (Pinwheel) | Acc (Swiss Roll)     | Latent NLL (Swiss Roll) | Acc (Square)        | Latent NLL (Square) |
> |-----------------|-------------------|-------------------|-------------------|-------------------|-------------------|-------------------|
> | VAE             | 0.16 ± 0.01     | 86.9 ± 99.03    | 0.15 ± 0.01     | 360.41 ± 185.87 | 0.15 ± 0.01     | 63.3 ± 70.64    |
> | IAF-VAE         | 0.19 ± 0.01     | 3.12 ± 1.16     | 0.16 ± 0.01     | 189.39 ± 57.48  | 0.18 ± 0.01     | 0.99 ± 0.11     |
> | AAE             | 0.23 ± 0.00     | 2.30 ± 0.07     | **0.23 ± 0.01** | **2.84 ± 0.08** | 0.20 ± 0.01     | 1.46 ± 0.49     |
> | HVAE w/ flow    | 0.20 ± 0.01     | 2.06 ± 0.07     | 0.20 ± 0.01     | 106.15 ± 15.30 | 0.19 ± 0.01     | 1.05 ± 0.06     |
> | DiffVAE         | **0.23 ± 0.01** | **1.37 ± 0.02** | **0.23 ± 0.01** | **2.80 ± 0.11** | **0.24 ± 0.01** | **0.86 ± 0.05** |
>
> [1] Ranganath, Rajesh, Dustin Tran, and David Blei. "Hierarchical variational models." International conference on machine learning. PMLR, 2016.
>
> [2] Vahdat, Arash, and Jan Kautz. "NVAE: A deep hierarchical variational autoencoder." Advances in neural information processing systems 33 (2020): 19667-19679.
>
> ***
> > **Q4** In Fig. 3 of the experiments, using a more complicated prior does not improve the visual quaility, and even doesn't see more diversity compared with AAE.
>
> **A4** We argue that for AAE, the last 8 samples look practically identical, even though they are sampled from different points along the prior. DiffVAE, on the other hand, has digits spread along the square much more evenly. AAE samples also miss the digit 4 and maybe 5 (the sixth sample from the left may be interpreted as the digit 5, but it also looks a lot like 0 and 8. DiffVAE, on the other hand, has a very nice digit 5.)

---

> ### Author Response · Authors · 2023-11-21
> **Response to Reviewer 3sxG (Part 3)**
>
> ***
> > **Q5** Can you point out the difference between DiffVAE and Variational diffusion models?
>
> **A5** In terms of the task they solve, VDM focuses on density estimation tasks where they achieve new state-of-the-art log-likelihoods using an introduced flexible family of diffusion-based generative models, while DiffVAE is motivated by **representation learning, dimensionality reduction and visualization tasks**, essentially tasks that VAEs are well-suited for, and seeks to improve variational inference via expressive posteriors based on diffusion-like framework.
>
> Content-wise, VDM focuses on improving the theoretical understanding of density modeling using diffusion models by analyzing the variational lower bound (VLB), deriving a remarkably simple expression in terms of the signal-to-noise ratio of the diffusion process.
>
> In contrast, DiffVAE focuses on enhancing VAE with diffusion-based models, where we augment variational inference in a latent variable model $p_\theta (\mathbf{x}, \mathbf{z})$ with auxiliary latent $\mathbf{y}$ introduced via a user-specified noising process$r(\mathbf{y} | \mathbf{z})$. Note that the diffusion models are used to instantiate the denoising variational inference framework. However, the transformation $r$ can be modeled using other approaches, e.g., discrete noising processes [1], and custom regularizers [2].
>
> [1] Austin, Jacob, et al. "Structured denoising diffusion models in discrete state-spaces." Advances in Neural Information Processing Systems 34 (2021): 17981-17993.
>
> [2] Miao, Ning, et al. "On incorporating inductive biases into VAEs." arXiv preprint arXiv:2106.13746 (2021).
>
> ***
> > **Q6** Some other methods like two stage VAE[1] are also learning a more complicated prior (without knowing the true density), then what is the strength of DiffVAE over these methods?
>
> **A6** Unlike DiffVAE, and other baselines in this paper, Two stage VAE does not allow enforcing a prior of the user's choice. Two stage VAE allows a mismatch between the aggregated posterior and the prior by training a second VAE that lets them sample from the aggregated posterior. Thus, it is more focused on learning a generative model with good sample quality than learning a generative model with a user-specified prior that might encode domain knowledge.
>
> ***
> > **Q7** Some other methods like two stage VAE[1] are also learning a more complicated prior (without knowing the true density), then what is the strength of DiffVAE over these methods?
>
> **A7**
>
> - Guarantees and Error Analysis as ELBO Variational Inference
>
> Since our DiffVAE lower bound operates within this same conceptual and mathematical framework, it inherits these guarantees of traditional ELBO in variational inference, whose convergence and error analysis are well-understood and extensively studied [1]. This means that the convergence behavior and error analysis of our DiffVAE lower bound can be understood and predicted with the same level of rigor as the ELBO in traditional variational inference.
>
> - Tightness of Lower Bound with Approximate Posterior Matching True Posterior
>
> In the principle of variational inference, an ELBO becomes tight, meaning it closely approximates the true log-likelihood, when the approximate posterior matches the true posterior of the data. In DiffVAE, we aim to achieve this tightness by using the denoising variational inference framework. As the model training progresses and the approximate posterior improves, the lower bound provided by our DiffVAE model becomes increasingly tight.
>
> [1] Damm, Simon, et al. "The ELBO of Variational Autoencoders Converges to a Sum of Entropies." International Conference on Artificial Intelligence and Statistics. PMLR, 2023.

---

> ### Author Response · Authors · 2023-11-21
> **Response to Reviewer 3sxG (Part 4)**
>
> ***
> > **Q8** What is the time consumption for the experiments? Compared with classical models?
>
> **A8** We conduct a computational cost analysis between the baselines and DiffVAE with various timesteps on the genotype clustering/visualization experiments. The table below shows that DiffVAE outperforms baselines at all timestamps and continues to improve after the baselines have plateaued.
>
> **Computational cost trade-off on 1kgenome: NMI vs wall-clock training time**
>
> NMI @ Wall-clock training time (min)
> | Method          | NMI @ 10 min | NMI @ 20 min | NMI @ 30 min | NMI @ 40 min| NMI @ 50 min| NMI @ 60 min|
> |-----------------|----------|----------|----------|----------|----------|----------|
> | VAE             | 0.52     | 0.52     | 0.52     | 0.52     | 0.52     | 0.52     |
> | IAF-VAE         | 0.54     | 0.52     | 0.52     | 0.52     | 0.52     | 0.52     |
> | AAE             | 0.61     | 0.57     | 0.57     | 0.57     | 0.57     | 0.57     |
> | DiffVAE (T=5)   | warm up  | 0.63     | 0.63     | 0.66     | 0.66     | 0.66     |
> | DiffVAE (T=10)  | warm up  | **0.64**     | **0.68**     | **0.70**     | **0.70**     | **0.70**     |
> | DiffVAE (T=20)  | warm up  | 0.50     | 0.51     | 0.56     | 0.64     | 0.68     |
> | DiffVAE (T=50)  | warm up  | 0.52     | 0.54     | 0.51     | 0.59     | 0.59     |

---

### Official Review · Reviewer_18s7 · 2023-10-31

**Soundness:** 2 fair
**Presentation:** 2 fair
**Contribution:** 2 fair
**Rating:** 5
**Confidence:** 4

**Summary:**

The paper proposes a hierarchical variational auto-encoder (VAE) with latents structured in Markov chain that is regularized to follow a diffusion process. To achieve this, authors augment the standard ELBO training objective with a de-noising regularization term, motivated by the "sleep" term of the wake-sleep algorithm. Authors evaluate the method on unsupervised learning, semi-supervised learning, and clustering+visualization, measuring the quality of the learned latent space.

**Strengths:**

Originality. The additional regularization term in Eq. (3) is novel (to the best of knowledge), and has a sound motivation: it allows us to prescribe a "forward" process for the sequence of latents.

**Weaknesses:**

Originality. While the regularization term in Eq. (3) is novel, the rest of the paper presents a standard hierarchical VAE. In particular, ladder VAEs by Sønderby et al. (2016) use a similar Markov chain of latents with Gaussian conditional step distributions. Subsequent work, including by Vahdat et al. (2020), further demonstrates the benefits of this approach. While implementation details might differ, the proposed method should be put in the context of this well-established area of research.

Significance. The method is compared to weak baselines, with the newest method (IAF-VAE) having been published in 2016. Stronger baselines should be used, including some of the hierarchical VAEs mentioned above. Unsupervised and semi-supervised experiments feel contrived: why do we care about these particular toy priors? Are there realistic settings in which we can demonstrate the method's benefits? The semi-supervised and clustering+visualization extensions and experiments feel secondary to the main contribution, and add little to the paper. It is not clear what the expected results in Figure 3 should be.

Clarity and Quality. Section 3 is very dense, and could be structured better. The value of Sections 3.2 and 3.3 is limited: their main messages could be summarized in the main text, with details moved to the appendix. The mathematical notation is hard to follow, and at times inconsistent: conditioning variables and parameter subscripts are dropped silently, variable names are mixed up (e.g. $\bar{H}$ in Eq. (6) is referred to as $H$ in the main text). Figure 1 (the main figure summarizing the method) has an error: $q_\phi(y|z,x)$ should be $q_\phi(z|y,x)$. This makes an already dense paper even harder to read.

## Additional references

- Sønderby, C. K., Raiko, T., Maaløe, L., Sønderby, S. K., & Winther, O. (2016). Ladder variational autoencoders. Advances in neural information processing systems, 29.
- Vahdat, A., & Kautz, J. (2020). NVAE: A deep hierarchical variational autoencoder. Advances in neural information processing systems, 33, 19667-19679.

**Questions:**

The paper mentions that the standard ELBO in Eq. (2) provides too weak of a learning signal. How does this look in practice? Have authors evaluated this approach on the benchmarks in the paper?

In what sense is the "latent space sleep term" in Eq. (7) an approximation of the sleep term in Eq. (3)? Authors mention that this regularization term has less computational overhead, but is an approximation, and hence the overall training objective is not a tight bound. How does this trade-off look in practice? How much computation are we saving, and how big of a sacrifice in terms of results are we seeing?

How does the computational complexity (or at least wall-clock training/inference time) of the method compare to the baselines?

---

> ### Author Response · Authors · 2023-11-21
> **Response to Reviewer 18s7 (Part 1)**
>
> We want to thank the reviewer for the thoughtful review and constructive feedback. We address each of your comments below:
>
> ***
> > **Q1** While the regularization term in Eq. (3) is novel, the rest of the paper presents a standard hierarchical VAE. While implementation details might differ, the proposed method should be put in the context of this well-established area of research. The method is compared to weak baselines, with the newest method (IAF-VAE) having been published in 2016. Stronger baselines should be used, including some of the hierarchical VAEs mentioned above.
>
> **A1** We would like to first note that we have established a connection between DiffVAE and auxiliary-variable VAE, a framework equivalent to hierarchical VAE, in section 3.2.
>
> To further strengthen this connection, we run a HVAE with normalizing flow [1]  baseline for unsupervised learning experiments and include a table comparing the training objectives and approximating family of different baselines with our method below:
>
> **Unsupervised learning on MNIST**
> | Method        | Acc (Pinwheel)      | Latent NLL (Pinwheel) | Acc (Swiss Roll)     | Latent NLL (Swiss Roll) | Acc (Square)        | Latent NLL (Square) |
> |---------------|---------------------|-----------------------|----------------------|-------------------------|---------------------|---------------------|
> | VAE           | 0.58 ± 0.01         | 6.18 ± 0.93           | 0.56 ± 0.055         | 82.96 ± 99.30           | 0.35 ± 0.02         | 25.07 ± 62.41       |
> | IAF-VAE       | **0.71 ± 0.00**     | 2.06 ± 0.21           | 0.67 ± 0.02          | 12.13 ± 18.28           | 0.65 ± 0.03         | 1.53 ± 0.82         |
> | AAE           | 0.67 ± 0.02         | 1.94 ± 0.09           | 0.68 ± 0.03          | 3.43 ± 0.14             | 0.57 ± 0.02         | 1.97 ± 0.70         |
> | HVAE w/ flow  | **0.72 ± 0.01**     | 2.18 ± 0.13           | **0.71 ± 0.02**      | 4.17 ± 1.23             | **0.70 ± 0.01**     | **1.25 ± 0.08**     |
> | DiffVAE       | **0.72 ± 0.01**     | **1.57 ± 0.10**       | 0.66 ± 0.01          | **3.19 ± 0.07**         | **0.72 ± 0.03**     | **1.24 ± 0.13**     |
>
> **Unsupervised learning on CIFAR-10**
> | Method        | Acc (Pinwheel)      | Latent NLL (Pinwheel) | Acc (Swiss Roll)     | Latent NLL (Swiss Roll) | Acc (Square)        | Latent NLL (Square) |
> |-----------------|-------------------|-------------------|-------------------|-------------------|-------------------|-------------------|
> | VAE             | 0.16 ± 0.01     | 86.9 ± 99.03    | 0.15 ± 0.01     | 360.41 ± 185.87 | 0.15 ± 0.01     | 63.3 ± 70.64    |
> | IAF-VAE         | 0.19 ± 0.01     | 3.12 ± 1.16     | 0.16 ± 0.01     | 189.39 ± 57.48  | 0.18 ± 0.01     | 0.99 ± 0.11     |
> | AAE             | 0.23 ± 0.00     | 2.30 ± 0.07     | **0.23 ± 0.01** | **2.84 ± 0.08** | 0.20 ± 0.01     | 1.46 ± 0.49     |
> | HVAE w/ flow    | 0.20 ± 0.01     | 2.06 ± 0.07     | 0.20 ± 0.01     | 106.15 ± 15.30 | 0.19 ± 0.01     | 1.05 ± 0.06     |
> | DiffVAE         | **0.23 ± 0.01** | **1.37 ± 0.02** | **0.23 ± 0.01** | **2.80 ± 0.11** | **0.24 ± 0.01** | **0.86 ± 0.05** |
>
> **Comparison of DiffVAE to other VAE-based methods**
> | Model                          | Training Objective     | Approximating Family             | Sampling-only Prior Compatibility
> |--------------------------------|------------------------|----------------------------------|-------------
> | VAE                            | ELBO                   | Diagonal Gaussian                | No |
> | IAF-VAE                        | ELBO                   | Normalizing flow                 | No |
> | AAE                            | Adversarial training   | Adversarial generator            | Yes |
> | HVAE w/o and w/ flow    | ELBO                   | Factorial Normal / Normalizing flow | No |
> | DiffVAE                        | ELBO + sleep term      | Denoising diffusion              | Yes |
>
> We note that we ran HVAE w/ flow instead of the newer NVAE [2] because the techniques introduced in NVAE are either orthogonal or inapplicable to our experiments. More details are below:
>
> NVAE made 4 contributions
>
> - Depthwise convolutions in its generative model – this is orthogonal as all VAE-based methods can use this
>
> - A new residual parameterization of the approximate posteriors – this is inapplicable as it requires certain forms of prior and do not allow the use of arbitrary prior of user choice
>
> - Spectral regularization – this is orthogonal as all VAE-based methods can use this
>
> - Practical solutions to reduce the memory burden of VAEs – this is orthogonal as all VAE-based methods can use this
>
> [1] Ranganath, Rajesh, Dustin Tran, and David Blei. "Hierarchical variational models." International conference on machine learning. PMLR, 2016.
>
> [2] Vahdat, Arash, and Jan Kautz. "NVAE: A deep hierarchical variational autoencoder." Advances in neural information processing systems 33 (2020): 19667-19679.

---

> ### Author Response · Authors · 2023-11-21
> **Response to Reviewer 18s7 (Part 2)**
>
> ***
> > **Q2** Unsupervised and semi-supervised experiments feel contrived: why do we care about these particular toy priors? Are there realistic settings in which we can demonstrate the method's benefits? The semi-supervised and clustering+visualization extensions and experiments feel secondary to the main contribution, and add little to the paper.
>
> The specific instantiation of the method used in this paper is motivated by representation learning, dimensionality reduction, and visualization tasks where expressive priors are used to encode domain knowledge. We conduct experiments on both synthetic and real benchmarks to support this application. Our MNIST/CIFAR-10 (standardly used in VAE evaluation [1]) experiments leverage widely-used priors [2, 3] and serve as synthetic benchmarks to evaluate the expressivity of the diffusion-based posterior. Then, we test diffusion variational inference on a real-world biological benchmark, 1000 genome, where DiffVAE outperforms all baselines. Semi-supervised learning and clustering experiments are performed to demonstrate that DiffVAE can be used in various visualization tasks: sometimes, data is partially labeled, and sometimes, the users only want to enforce “soft” priors, such as mixture of Gaussian, on the visualization.
>
> [1] Vahdat, Arash, and Jan Kautz. "NVAE: A deep hierarchical variational autoencoder." Advances in neural information processing systems 33 (2020): 19667-19679.
>
> [2] Alireza Makhzani, Jonathon Shlens, Navdeep Jaitly, Ian Goodfellow, and Brendan Frey. Adversarial autoencoders. arXiv preprint arXiv:1511.05644, 2015
>
> [3] Matthew J. Johnson, David Duvenaud, Alexander B. Wiltschko, Sandeep Robert Datta, and Ryan P. Adams. Composing graphical models with neural networks for structured representations and fast inference. In Advances in Neural Information Processing Systems (NIPS) 29, 2016. arXiv:1603.06277 [stat.ML].
>
>
> ***
> > **Q3** It is not clear what the expected results in Figure 3 should be.
>
> **A3** We argue that for AAE, the last 8 samples look practically identical, even though they are sampled from different points along the prior. DiffVAE, on the other hand, has digits spread along the square much more evenly. AAE samples also miss the digit 4 and maybe 5 (the sixth sample from the left may be interpreted as the digit 5, but it also looks a lot like 0 and 8. DiffVAE, on the other hand, has a very nice digit 5.)
>
> ***
> > **Q4** The value of Sections 3.2 and 3.3 is limited. The mathematical notation is hard to follow, and at times inconsistent
>
> **A4** We thank the reviewer for identifying the typos, which we have corrected. We have also moved section 3.2 and 3.3 to the appendix for better readability.
>
> ***
> > **Q5** The paper mentions that the standard ELBO in Eq. (2) provides too weak of a learning signal. How does this look in practice? Have authors evaluated this approach on the benchmarks in the paper?
>
> **A5** We show results of VAE with diffusion encoder trained with standard ELBO below:
>
> **Unsupervised learning on MNIST, including the failed results of VAE with Diffusion**
> |    Method              | Latent NLL  (Pinwheel)       | Latent NLL  (Swiss Roll)      | Latent NLL  (Square)      |
> | --------------- | ------------------ | ------------------ | ----------------- |
> | VAE             | 6.18 ± 0.93        | 82.96 ± 99.30      | 25.07 ± 62.41     |
> | IAF-VAE         | 2.06 ± 0.21        | 12.13 ± 18.28      | 1.53 ± 0.82       |
> | AAE             | 1.94 ± 0.09        | 3.43 ± 0.14        | 1.97 ± 0.70       |
> | HVAE w/ flow    | 2.18 ± 0.13        | 4.17 ± 1.23        | **1.25 ± 0.08**   |
> | DiffVAE         | **1.57 ± 0.10**    | **3.19 ± 0.07**    | **1.24 ± 0.13**   |
> | **VAE with Diffusion (standard ELBO)** | 6.09 | 510.53 | 265.25 |
>
> As you can see, VAE with Diffusion (standard ELBO) totally fails to respect the prior. The intuition is as follows:
>
> The objective samples from the reverse diffusion process instead of the forward diffusion process like in standard diffusion training (please see Sec. 3.1.1 for the details). Specifically, for the term
> $E_{q_\psi(\mathbf{z_{0:T}}|\mathbf{x})} [D_{KL}(q_\psi(\mathbf{z_{t-1}}|\mathbf{z_t}, \mathbf{x})||r(\mathbf{z_{t-1}}|\mathbf{z_t}, \mathbf{z_0}))]$ in VAE with Diffusion (standard ELBO) objective,
> we have to backpropagate through $\mathbf{z_0}$ and $\mathbf{z_t}$ all the way to $\mathbf{z_T}$. This long chain of backpropagation makes the learning signal weak; the same does not happen in standard diffusion training because $z_0$ is the data that does not need to be backpropagated.

---

> ### Author Response · Authors · 2023-11-21
> **Response to Reviewer 18s7 (Part 3)**
>
> ***
> > **Q6** In what sense is the "latent space sleep term" in Eq. (7) an approximation of the sleep term in Eq. (3)?
>
> **A6** The "latent space sleep term" in Equation (7) serves as an approximation of the sleep term in Equation (3) primarily because it bypasses the need to sample from $p_\theta(\mathbf{x})$, instead focusing solely on sampling the latent variables. This modification leads to a less tight ELBO, as discussed in Sec. 3.1.2. The key benefit of this approach is the increased computational efficiency gained from not having to sample $p_\theta(\mathbf{x})$, making the process faster. Overall, the approximation is designed to retain the effectiveness of the original term while making the computational process more feasible.
>
> ***
> > **Q7** Authors mention that this regularization term has less computational overhead, but is an approximation, and hence the overall training objective is not a tight bound. How does this trade-off look in practice? How does the computational complexity (or at least wall-clock training/inference time) of the method compare to the baselines?
>
> **A7** We conduct a computational cost analysis between the baselines and DiffVAE with various timesteps on the genotype clustering/visualization experiments. The table below shows that DiffVAE outperforms baselines at all timestamps and continues to improve after the baselines have plateaued.
>
> **Computational cost trade-off on 1kgenome: NMI vs wall-clock training time**
>
> NMI @ Wall-clock training time (min)
> | Method          | NMI @ 10 min | NMI @ 20 min | NMI @ 30 min | NMI @ 40 min| NMI @ 50 min| NMI @ 60 min|
> |-----------------|----------|----------|----------|----------|----------|----------|
> | VAE             | 0.52     | 0.52     | 0.52     | 0.52     | 0.52     | 0.52     |
> | IAF-VAE         | 0.54     | 0.52     | 0.52     | 0.52     | 0.52     | 0.52     |
> | AAE             | 0.61     | 0.57     | 0.57     | 0.57     | 0.57     | 0.57     |
> | DiffVAE (T=5)   | warm up  | 0.63     | 0.63     | 0.66     | 0.66     | 0.66     |
> | DiffVAE (T=10)  | warm up  | **0.64**     | **0.68**     | **0.70**     | **0.70**     | **0.70**     |
> | DiffVAE (T=20)  | warm up  | 0.50     | 0.51     | 0.56     | 0.64     | 0.68     |
> | DiffVAE (T=50)  | warm up  | 0.52     | 0.54     | 0.51     | 0.59     | 0.59     |

---

> ### Comment · Reviewer_18s7 · 2023-11-22
>
> I thank the authors for their clarifications and additional results. Some of my points have been addressed, so I increase my score.
>
> At the same time, the story is still not entirely clear to me, and the results above raise further questions. In particular:
> - We could theoretically use the sleep term with the factorial normal / normalizing flow priors -- could it be responsible for the improved results of DiffVAE, or is it indeed the combination of the sleep term + the diffusion prior? (Your additional results show that it's not the diffusion prior by itself, at least.)
> - The interpretation of the Figure 3 is not entirely convincing to me, partially due to the overall low quality of generations.
> - The answer to Q6 is also not entirely satisfying: I do understand that the approximation is cheaper at a cost of a looser ELBO, but it would be useful to better characterize this trade-off, either theoretically or empirically.
>
> Overall, I think the paper contains a set of intriguing ideas, but could do a better job tying these ideas into a coherent narrative that is well placed in the context of prior work, and digging deeper into core components of the method.

---

### Official Review · Reviewer_t61r · 2023-10-31

**Soundness:** 2 fair
**Presentation:** 2 fair
**Contribution:** 2 fair
**Rating:** 5
**Confidence:** 3

**Summary:**

This paper studies Denoising Diffusion Variational Inference - using diffusion models, a class of generative models known for high-quality sample generation, to fit a complex posterior through a diffusion process in the latent space. The approach involves augmenting a variational posterior with auxiliary latent variables introduced through a noising process $r$, which transforms a complex latent $z$ to a simple auxiliary latent $y$. The noising process r diffuses z into y through multiple steps, as in a denoising diffusion model. r is set to be a Gaussian diffusion but can be set to another stochastic process. To perform inference in the augmented latent variable model, an approximate posterior $q(y|z)$ is fit along with the forward (denoising) process $q(z|x,y)$. The learning objective is based on the wake-sleep algorithm, and it essentially the ELBO with an additional term which acts as regularization. The authors consider an instantiation of this framework in the context of VAEs, termed DiffVAE, leveraging this flexible encoder $q$. The approach is evaluated empirically in dimensionality reduction and representation learning tasks, outperforming methods based on adversarial training or invertible flow-based posteriors. Specifically. the authors consider a biology task—inferring latent ancestry from human genomes—and demonstrate that DiffVAE outperforms strong baselines on the 1000 Genomes dataset.

**Strengths:**

* Using a denoising diffusion model as an encoder enables modelling of complex posteriors leveraging the expressive power of diffusion models. This can be useful for representation learning tasks where we want latent space to match semantic structure which can require the encoder to express highly structured distributions.

* By relying on a wake-sleep style approach, the method avoids adversarial training and requirements of constrained architectures like flows to enabling flexible modeling.

* The noising process is a key component of the approach. While in the paper a Gaussian diffusion is used, the noising process can also provide a way to impose some structure based on prior knowledge.

* The method shows good empirical performance in semi-supervised tasks with label-conditional priors and genotype clustering tasks.

**Weaknesses:**

* The paper presents the very general framework of using diffusion models to model expressive posteriors for variational inference. However, the specific instantiations studied here - semi-supervised learning and clustering - seem quite a bit limited. My main concern is about the generality of the approach. While in principle the approach is applicable to various LVM problems the experiments only study two specific problems.
* Moreover, the tasks considered here seem relatively simple, not enough to demonstrate the effect of the increased expressivity.
* The increased expressivity also comes at a computational cost - sampling from the diffusion model can be quite expensive (for example in the experiments the authors use 20 and 100 steps which means 20 forward passes of a neural network.
* The method also introduces a critical new hyperparameter - the number of steps for diffusion. The paper does not discuss how this parameter is selected and how sensitive the results are to the choice of this parameter.
* Reproducibility: The authors do not include code to reproduce their results but most details seem to be included in the paper.

**Questions:**

* What is the effect of the number of steps of the diffusion process on the performance?
* Where do the authors believe would the increased expressivity would be useful enough to trade-off the computation cost?

---

> ### Author Response · Authors · 2023-11-21
> **Response to Reviewer t61r (Part 1)**
>
> We want to thank the reviewer for the thoughtful review and constructive feedback. We address each of your comments below:
>
> ***
> > **Q1** However, the specific instantiations studied here - semi-supervised learning and clustering - seem quite a bit limited. My main concern is about the generality of the approach. Moreover, the tasks considered here seem relatively simple, not enough to demonstrate the effect of the increased expressivity.
>
> **A1** The specific instantiation of the method used in this paper is motivated by representation learning, dimensionality reduction, and visualization tasks where expressive priors are used to encode domain knowledge. We conduct experiments on both synthetic and real benchmarks to support this application. Our MNIST/CIFAR-10 (standardly used in VAE evaluation [1]) experiments leverage widely-used priors [2, 3] and serve as synthetic benchmarks to evaluate the expressivity of the diffusion-based posterior. Then, we test diffusion variational inference on a real-world biological benchmark, 1000 genome, where DiffVAE outperforms all baselines. Semi-supervised learning and clustering experiments are performed to demonstrate that DiffVAE can be used in various visualization tasks: sometimes, data is partially labeled, and sometimes, the users only want to enforce “soft” priors, such as mixture of Gaussian, on the visualization.
>
> [1] Vahdat, Arash, and Jan Kautz. "NVAE: A deep hierarchical variational autoencoder." Advances in neural information processing systems 33 (2020): 19667-19679.
>
> [2] Alireza Makhzani, Jonathon Shlens, Navdeep Jaitly, Ian Goodfellow, and Brendan Frey. Adversarial autoencoders. arXiv preprint arXiv:1511.05644, 2015
>
> [3] Matthew J. Johnson, David Duvenaud, Alexander B. Wiltschko, Sandeep Robert Datta, and Ryan P. Adams. Composing graphical models with neural networks for structured representations and fast inference. In Advances in Neural Information Processing Systems (NIPS) 29, 2016. arXiv:1603.06277 [stat.ML].
>
> ***
> > **Q2** The increased expressivity also comes at a computational cost - sampling from the diffusion model can be quite expensive (for example in the experiments the authors use 20 and 100 steps which means 20 forward passes of a neural network. Where do the authors believe the increased expressivity would be useful enough to trade-off the computation cost?
>
> **A2** We would like to point out that we are conducting a diffusion process with low-dimensional latents, which means the computational overhead will not be affected much. E.g., for MNIST, the dimensionality (1xd) of latents is much smaller than the actual data (28x28) size.
>
> To further support the efficiency of DiffVAE, we conduct a computational cost analysis between the baselines and DiffVAE with various timesteps on the genotype clustering/visualization experiments. The table below shows that DiffVAE outperforms baselines at all timestamps and continues to improve after the baselines have plateaued.
>
> **Computational cost trade-off on 1kgenome: NMI vs wall-clock training time**
>
> NMI @ Wall-clock training time (min)
> | Method          | NMI @ 10 min | NMI @ 20 min | NMI @ 30 min | NMI @ 40 min| NMI @ 50 min| NMI @ 60 min|
> |-----------------|----------|----------|----------|----------|----------|----------|
> | VAE             | 0.52     | 0.52     | 0.52     | 0.52     | 0.52     | 0.52     |
> | IAF-VAE         | 0.54     | 0.52     | 0.52     | 0.52     | 0.52     | 0.52     |
> | AAE             | 0.61     | 0.57     | 0.57     | 0.57     | 0.57     | 0.57     |
> | DiffVAE (T=5)   | warm up  | 0.63     | 0.63     | 0.66     | 0.66     | 0.66     |
> | DiffVAE (T=10)  | warm up  | **0.64**     | **0.68**     | **0.70**     | **0.70**     | **0.70**     |
> | DiffVAE (T=20)  | warm up  | 0.50     | 0.51     | 0.56     | 0.64     | 0.68     |
> | DiffVAE (T=50)  | warm up  | 0.52     | 0.54     | 0.51     | 0.59     | 0.59     |
>
> Since our key application here is dimensionality reduction and visualization, all experiments in our paper run in just a few hours max, not days. We believe that the additional training time is still reasonable such that practitioners are willing to wait for better results.

---

> ### Author Response · Authors · 2023-11-21
> **Response to Reviewer t61r (Part 2)**
>
> ***
> > **Q3** The method also introduces a critical new hyperparameter - the number of steps for diffusion. The paper does not discuss how this parameter is selected and how sensitive the results are to the choice of this parameter.
>
> **A3** We use 20 timesteps as the default setup of this hyperparameter. And we keep it unchanged across all experiments in our paper. The results show that with this default value, we can match the aggregated posterior and the prior well, which indicates it is, at least, not too sensitive that needs exhaustive tuning.
>
> The users of the method can also inspect the visualization results and increase it if there is a relatively big mismatch between the aggregated posterior and the prior, as we did for semi-supervised learning experiments. The key here is that the number of steps for diffusion is the type of hyperparameter that is the higher the better, at the expense of longer training time. Thus, it is not as hard to tune as other types of hyperparameters where making them too low or too high can be both bad.
>
> To support this, we’re copying results from computational cost analysis from part 2 of the replies that shows the performance of DiffVAE at different timesteps here:
>
> NMI @ Wall-clock training time (min)
> | Method          | NMI @ 10 min | NMI @ 20 min | NMI @ 30 min | NMI @ 40 min| NMI @ 50 min| NMI @ 60 min|
> |-----------------|----------|----------|----------|----------|----------|----------|
> | DiffVAE (T=5)   | warm up  | 0.63     | 0.63     | 0.66     | 0.66     | 0.66     |
> | DiffVAE (T=10)  | warm up  | **0.64**     | **0.68**     | **0.70**     | **0.70**     | **0.70**     |
> | DiffVAE (T=20)  | warm up  | 0.50     | 0.51     | 0.56     | 0.64     | 0.68     |
> | DiffVAE (T=50)  | warm up  | 0.52     | 0.54     | 0.51     | 0.59     | 0.59     |
>
> We note that the model with T=10 outperforms the one with T=5, and the performances of the models with T=20 and T=50 have not yet plateaued at 60 min.

---

> ### Comment · Reviewer_t61r · 2023-11-22
>
> Thanks for the response!
>
> A1: I feel I have to push back on that a bit. As you mentioned CIFAR/MNIST are synthetic benchmarks but seem insufficient for a method where added expressiveness is a major pitch. The biological task is indeed interesting but it is hard to understand how well that performance transfers to other domains.
>
> A2: Thanks for the runtime details, it is quite helpful. It does seem like the results are not sensitive to the diffusion steps. However, this potentially points to my previous concerns of the tasks being "too easy" to take advantage of the benefits afforded by DiffVAE.
>
> Overall, I agree with Reviewer 18s7 that the paper has an intriguing set of ideas but falls short of demonstrating the effectiveness on appropriate tasks and benchmarks.

---

> > ### Author Response · Authors · 2023-11-22
> > **Task Clarification**
> >
> > We thank the reviewer for the quick reply, and we would like to note that while we have not explored tasks like density estimation or high-quality sample generation in this paper, the specific instantiation of the framework used in this paper is well-motivated by biological data visualization tasks, of which there are immediate practical values and interests as explained in the paper. The experiments on MNIST and CIFAR-10 with the priors used in the paper are done specifically to support the potentials of the method at biological data visualization tasks. Then, we show strong performance on the realistic 1000 genomes dataset. We emphasize the practical values of the instantiation used in the paper for biological data visualization tasks.

---

### Official Review · Reviewer_Qkoy · 2023-11-01

**Soundness:** 3 good
**Presentation:** 3 good
**Contribution:** 2 fair
**Rating:** 5
**Confidence:** 4

**Summary:**

This paper proposes a novel latent variable inference method, DiffVAE that combines the variational autoencoder and diffusion models. More specifically, DiffVAE enhances the expressive power of the variational posterior by designing a denoising process on augmented latent variables which transforms a complex latent variable into a simple augmented latent variables. Alternative perspective from auxiliary variable models and regularization, and extensions to semi-supervised learning and the clustering are also provided. In experiments, the authors demonstrate the effectiveness of their method on both unsupervised learning, semi-supervised learning genotype clustering tasks. The results indicate that DiffVAE performs better in terms of NLL metric and clustering quality in the latent space.

**Strengths:**

- The paper proposes a novel method to enhance the expressive power of the variational posterior in VAE by diffusion model.
- The adaption of wake-sleep algorithm for a more informative training of the encoder is interesting.
- The paper is written clearly, especially on related background and alternative perspectives.

**Weaknesses:**

The DiffVAE method is natural and interesting. However, the method is closely related to auxiliary variable models/hierarchical variational models (HVM) [1] (albeit some small modifications such as additional regularizations). Also, the design of the training objective needs better justification.

1. The authors propose $\hat{\mathcal{L}}$ in equation (7) to replace the original lower bound $\mathcal{L}$ in order to address the requirement of sampling $x$ from $p_\theta$.
They consider $L_{\mathrm{sleep}}(x,\phi)$ as a regularization of the original ELBO.
I'm concerned that this part of the process may lack sufficient justification since the gradient of $q_\phi(z|x)$ is biased for the original ELBO.
2. There aren't much detail about the updating scheme of parameters $\phi$ and $\theta$ in the experiments.

[1] Rajesh Ranganath, Dustin Tran, and David Blei. Hierarchical variational models. In International conference on machine learning, pp. 324–333. PMLR, 2016.

**Questions:**

- In section 3.1.1., "learning signal from this procedure to be too weak to learn a good $q_\phi(z|y)$" is not clear enough, can you clarify this point more clearly? This is a crucial point that motivates your approach (otherwise, HVM would be enough)
- How are the parameters $\theta$ and $\phi$ updated? Is $\theta$ only updated from the reconstruction loss and $\phi$ only updated from the sleep loss?
- It seems that the $\mathcal{L}_{\mathrm{sleep}}$ term does nothing but pushing the encoder towards the prior. It is, therefore, no wonder that DiffVAE performs better in term of recovering the true prior. It would be better to compare the NLL (instead of latent NLL) for the learned generative model as well.
- The paper would be strengthened by including HVM with a fixed reversed model as a baseline model (which is equivalent to DiffVAE without $\mathcal{L}_{\mathrm{sleep}}$) in the experiments.
- As a method mentioned in the related work, [1] proposed LSGM, which can also be seen as a VAE with a reversed diffusion prior. Could you please provide a more detailed comparison between this method and DiffVAE, and clarify whether it can also be used for semi-supervised learning?

Typos

- In the third paragraph of the background section, the term $ D_{\mathrm{KL}}(p_\theta(z|x)||q_\phi(z|x)$ should be $ D_{\mathrm{KL}}(p_\theta(z|x)||q_\phi(z|x))$.
- Omissions that may lead to misunderstandings. For example, the expectation in equation (4) should be $p_\theta(x,z)$ rather than $p(x,z)$; $D_{KL}(p_\theta\|q(z|x))$ should be $D_{KL}(p_\theta\|q_\phi(z|x))$.
- The second row corresponding to the prior appears to be square rather than Swissroll in figure 2.

[1] Vahdat, Arash, Karsten Kreis, and Jan Kautz. "Score-based generative modeling in latent space." NIPS 2021.

---

> ### Author Response · Authors · 2023-11-21
> **Response to Reviewer Qkoy (Part 1)**
>
> We want to thank the reviewer for the thoughtful review and constructive feedback. We address each of your comments below:
>
> ***
> > **Q1** However, the method is closely related to auxiliary variable models/hierarchical variational models (HVM) [1] (albeit some small modifications such as additional regularizations).
> Also, the design of the training objective needs better justification.
> In section 3.1.1., "learning signal from this procedure to be too weak to learn a good q(z|y) is not clear enough, can you clarify this point more clearly?
>
> **A1** We would like to include here a table comparing the training objectives and approximating family of different baselines, including a newly added HVAE w/ flow baseline, with our method below:
>
> **Comparison of DiffVAE to other relevant methods**
> | Model   | Training Objective | Approximating Family | Sample-based Prior | Auxiliary Variable | Tasks | Simplified Graphical Illustration |
> |---------|--------------------|----------------------|--------------------|--------------------|-------|-----------------------------------|
> | VAE 	| ELBO           	| Diagonal Gaussian	| No             	| No             	| Density estimation | $\mathbf{x}$ -> $\mathbf{z}$ -> $\mathbf{x}$ |
> | IAF-VAE | ELBO           	| Normalizing flow 	| No             	| Yes            	| Density estimation/Visualization | $\mathbf{x}$ -> $\mathbf{z}_0$ -> $\mathbf{z}_T$ -> $\mathbf{x}$ |
> | AAE 	| Adversarial training | Adversarial generator | Yes           	| No             	| Visualization | $\mathbf{x}$ -> $\mathbf{z}$ -> $\mathbf{x}$ |
> | HVAE w/o and w/ flow | ELBO | Factorial Normal / Normalizing flow | No 	| Yes           	| Density estimation/High-quality sample generation | $\mathbf{x}$ -> $\mathbf{z}_0$ -> $\mathbf{z}_T$ -> $\mathbf{z}_0$ -> $\mathbf{x}$ |
> | ADGM	| ELBO           	| Non-Gaussian     	| No             	| Yes            	| Density estimation | $\mathbf{x}$ -> $\mathbf{a}$ -> $\mathbf{z}$ -> $\mathbf{x}$ |
> | LDM 	| ELBO           	| Diagonal Gaussian	| No             	| Yes            	| High-quality sample generation | $\mathbf{x}$ -> $\mathbf{z}_0$ -> $\mathbf{z}_T$ -> $\mathbf{z}_0$ -> $\mathbf{x}$ |
> | LSGM	| ELBO+score matching | Diagonal Gaussian   | No             	| Yes            	| High-quality sample generation | $\mathbf{x}$ -> $\mathbf{z}_0$ -> $\mathbf{z}_T$ -> $\mathbf{z}_0$ -> $\mathbf{x}$ |
> | DiffVAE | ELBO+sleep term	| Denoising diffusion | Yes            	| Yes            	| Density estimation/Visualization | $\mathbf{x}$ -> $\mathbf{z}_T (\mathbf{y})$ -> $\mathbf{z}_0 (\mathbf{z})$ -> $\mathbf{x}$ |
>
> To justify the need for our training objective, we show (failed) results of VAE with diffusion encoder trained with standard ELBO below:
>
> **Unsupervised learning on MNIST, including the failed results of VAE with Diffusion**
> |    Method              | Latent NLL  (Pinwheel)       | Latent NLL  (Swiss Roll)      | Latent NLL  (Square)      |
> | --------------- | ------------------ | ------------------ | ----------------- |
> | VAE             | 6.18 ± 0.93        | 82.96 ± 99.30      | 25.07 ± 62.41     |
> | IAF-VAE         | 2.06 ± 0.21        | 12.13 ± 18.28      | 1.53 ± 0.82       |
> | AAE             | 1.94 ± 0.09        | 3.43 ± 0.14        | 1.97 ± 0.70       |
> | HVAE w/ flow    | 2.18 ± 0.13        | 4.17 ± 1.23        | **1.25 ± 0.08**   |
> | DiffVAE         | **1.57 ± 0.10**    | **3.19 ± 0.07**    | **1.24 ± 0.13**   |
> | **VAE with Diffusion (standard ELBO)** | 6.09 | 510.53 | 265.25 |
>
> As you can see, VAE with Diffusion (standard ELBO) totally fails to respect the prior. The intuition is as follows:
>
> The objective samples from the reverse diffusion process instead of the forward diffusion process like in standard diffusion training (please see Sec. 3.1.1 for the details). Specifically, for the term
> $E_{q_\psi(\mathbf{z_{0:T}}|\mathbf{x})} [D_{KL}(q_\psi(\mathbf{z_{t-1}}|\mathbf{z_t}, \mathbf{x})||r(\mathbf{z_{t-1}}|\mathbf{z_t}, \mathbf{z_0}))]$ in VAE with Diffusion (standard ELBO) objective,
> we have to backpropagate through $\mathbf{z_0}$ and $\mathbf{z_t}$ all the way to $\mathbf{z_T}$. This long chain of backpropagation makes the learning signal weak; the same does not happen in standard diffusion training because $z_0$ is the data that does not need to be backpropagated.
>
> ***
> > **Q2** They consider L_sleep as a regularization of the original ELBO. I'm concerned that this part of the process may lack sufficient justification since the gradient of q(z|x) is biased for the original ELBO.
>
> **A2** The concern regarding the bias in the gradient of $q (\mathbf{z} | \mathbf{x})$ for the original ELBO is acknowledged. However, in DiffVAE, we've introduced a new lower bound specifically designed for our model. The gradient is unbiased for our objective, which is a valid lower bound; empirically, this new lower bound works better.

---

> ### Author Response · Authors · 2023-11-21
> **Response to Reviewer Qkoy (Part 2)**
>
> ***
> > **Q3** There aren't much detail about the updating scheme of parameters? How are the parameters updated?
>
> **A3** Here is the pseudocode for the training algorithm
>
> ***
> DiffVAE Training Algorithm
> ---
> For each batch of inputs $\mathbf{X}$
>
> Optimize $\theta$ and $\phi$ with respect to $\mathcal{L_{rec}}(\mathbf{X}, \theta, \phi) + \beta \cdot \mathcal{L_{reg}}(\mathbf{X}, \theta, \phi)$
>
> Optimize $\phi$ with respect to $\gamma \cdot \mathcal{L_{sleep}}(\mathbf{X}, \phi)$
>
> End
> ***
>
> The training algorithm alternates between optimizing $\theta$ and $\phi$ with respect to $\mathcal{L_{rec}} + \beta \cdot \mathcal{L_{reg}}$ and optimizing only $\phi$ with respect to $\mathcal{L_{sleep}}$
>
> ***
> > **Q4** The paper would be strengthened by including HVM with a fixed reversed model as a baseline model (which is equivalent to DiffVAE without L_sleep in the experiments.
>
> **A4** We run a HVAE with normalizing flow [1]  baseline for unsupervised learning experiments, please see results below:
>
> **Unsupervised learning on MNIST**
> | Method        | Acc (Pinwheel)      | Latent NLL (Pinwheel) | Acc (Swiss Roll)     | Latent NLL (Swiss Roll) | Acc (Square)        | Latent NLL (Square) |
> |---------------|---------------------|-----------------------|----------------------|-------------------------|---------------------|---------------------|
> | VAE           | 0.58 ± 0.01         | 6.18 ± 0.93           | 0.56 ± 0.055         | 82.96 ± 99.30           | 0.35 ± 0.02         | 25.07 ± 62.41       |
> | IAF-VAE       | **0.71 ± 0.00**     | 2.06 ± 0.21           | 0.67 ± 0.02          | 12.13 ± 18.28           | 0.65 ± 0.03         | 1.53 ± 0.82         |
> | AAE           | 0.67 ± 0.02         | 1.94 ± 0.09           | 0.68 ± 0.03          | 3.43 ± 0.14             | 0.57 ± 0.02         | 1.97 ± 0.70         |
> | HVAE w/ flow  | **0.72 ± 0.01**     | 2.18 ± 0.13           | **0.71 ± 0.02**      | 4.17 ± 1.23             | **0.70 ± 0.01**     | **1.25 ± 0.08**     |
> | DiffVAE       | **0.72 ± 0.01**     | **1.57 ± 0.10**       | 0.66 ± 0.01          | **3.19 ± 0.07**         | **0.72 ± 0.03**     | **1.24 ± 0.13**     |
>
> **Unsupervised learning on CIFAR-10**
> | Method        | Acc (Pinwheel)      | Latent NLL (Pinwheel) | Acc (Swiss Roll)     | Latent NLL (Swiss Roll) | Acc (Square)        | Latent NLL (Square) |
> |-----------------|-------------------|-------------------|-------------------|-------------------|-------------------|-------------------|
> | VAE             | 0.16 ± 0.01     | 86.9 ± 99.03    | 0.15 ± 0.01     | 360.41 ± 185.87 | 0.15 ± 0.01     | 63.3 ± 70.64    |
> | IAF-VAE         | 0.19 ± 0.01     | 3.12 ± 1.16     | 0.16 ± 0.01     | 189.39 ± 57.48  | 0.18 ± 0.01     | 0.99 ± 0.11     |
> | AAE             | 0.23 ± 0.00     | 2.30 ± 0.07     | **0.23 ± 0.01** | **2.84 ± 0.08** | 0.20 ± 0.01     | 1.46 ± 0.49     |
> | HVAE w/ flow    | 0.20 ± 0.01     | 2.06 ± 0.07     | 0.20 ± 0.01     | 106.15 ± 15.30 | 0.19 ± 0.01     | 1.05 ± 0.06     |
> | DiffVAE         | **0.23 ± 0.01** | **1.37 ± 0.02** | **0.23 ± 0.01** | **2.80 ± 0.11** | **0.24 ± 0.01** | **0.86 ± 0.05** |
>
> [1] Ranganath, Rajesh, Dustin Tran, and David Blei. "Hierarchical variational models." International conference on machine learning. PMLR, 2016.
>
> ***
> > **Q5** As a method mentioned in the related work, [1] proposed LSGM, which can also be seen as a VAE with a reversed diffusion prior. Could you please provide a more detailed comparison between this method and DiffVAE, and clarify whether it can also be used for semi-supervised learning?
>
> **A5** We have a discussion of it in our paper. We copy it here for convenience: “These works focus on sample quality and directly fit $q_\phi (\mathbf{z} | \mathbf{x})$ to a complex distribution; our work instead seeks to achieve high sample quality in latent space, and obtains complex $\mathbf{z}$ by diffusion from a simple $\mathbf{y}  \sim q_\phi(\mathbf{y} | \mathbf{x})$.”
>
> To provide more details, contrasting our model with Latent Diffusion Models (LDM) and Latent Score-based Generative Models (LSGM), a key difference lies in the nature and role of the latent variables. In LDM and LSGM, the latent variables typically represent downscaled images, which the decoder then upscales to generate the final output. In our model, however, the latent variables have a different interpretation. They are the auxiliary variables of the encoder that make the approximate posterior more expressive.
>
> Both can be used for semi-supervised learning.
>
> ***
> > **Q6** Typos
>
> **A6** We thank the reviewer for identifying the typos, which we have corrected

---

> > ### Comment · Reviewer_Qkoy · 2023-11-22
> >
> > Thank you for your response. I would like to keep my score.

---

> > > ### Author Response · Authors · 2023-11-23
> > > **Response by Authors**
> > >
> > > We thank the reviewer for the quick reply. Please feel free to let us know if you have any other concerns and if there is anything we can address to potentially improve our score.

---

### Author Response · Authors · 2023-11-22
**General response to all reviewers**

We thank all the reviewers for their detailed feedback. We have posted responses to each reviewer's individual comments. Here, we detail all the changes that we have made to our manuscript in light of this rebuttal period:
- We ran a new HVAE w/ flow baseline for unsupervised learning experiments on MNIST and CIFAR-10 and show that DiffVAE still outperforms it
- We conducted a computational cost analysis experiment and show that DiffVAE outperforms baselines at all timestamps and continues to improve after the baselines have plateaued
- We add a new empirical evidence showing the failed results of VAE with Diffusion encoder (standard ELBO) to further justify the introduction of our learning objectives
- We clarified differences between different VAE-based methods and our method with a table displaying different attributes of each method
- We fixed the typos pointed out by the reviewers

One common concern reviewers have is the experimental setup used in the paper, that it might be too simple. We would like to note that while we have not explored tasks like density estimation or high-quality sample generation in this paper, the specific instantiation of the framework used in this paper is well-motivated by biological data visualization tasks, of which there are immediate practical values and interests as explained in the paper. The experiments on MNIST and CIFAR-10 with the priors used in the paper are done specifically to support the potentials of the method at biological data visualization tasks. Then, we show strong performance on the realistic 1000 genomes dataset. We emphasize the practical values of the instantiation used in the paper for biological data visualization tasks.

---

### Meta-Review · Area_Chair_eehd · 2023-12-08

**Metareview:**

Summary:

The paper proposes a hierarchical variational auto-encoder (VAE) with latents structured in Markov chain that is regularized to follow a diffusion process. To achieve this, authors augment the standard ELBO training objective with a de-noising regularization term, motivated by the "sleep" term of the wake-sleep algorithm. Authors evaluate the method on unsupervised learning, semi-supervised learning, and clustering+visualization, measuring the quality of the learned latent space. In experiments, the authors demonstrate the effectiveness of their method on both unsupervised learning, semi-supervised learning genotype clustering tasks. The results indicate that DiffVAE performs better in terms of NLL metric and clustering quality in the latent space.

Strengths:

- The paper proposes a novel method to enhance the expressive power of the variational posterior in VAE by diffusion model.
- The adaption of wake-sleep algorithm for a more informative training of the encoder is interesting.
- The paper is written clearly, especially on related background and alternative perspectives.
- Using a denoising diffusion model as an encoder enables modelling of complex posteriors leveraging the expressive power of diffusion models.
- By relying on a wake-sleep style approach, the method avoids adversarial training and requirements of constrained architectures like flows to enabling flexible modeling.
- The noising process is a key component of the approach. While in the paper a Gaussian diffusion is used, the noising process can also provide a way to impose some structure based on prior knowledge.
- The method shows good empirical performance in semi-supervised tasks with label-conditional priors and genotype clustering tasks.
- The additional regularization term in Eq. (3) is novel and has a sound motivation.

Weaknesses:

- The method is closely related to auxiliary variable models/hierarchical variational models (HVM).
- The design of the training objective needs better justification.
- There aren't much detail about the updating scheme of parameters and in the experiments.
- The specific instantiations studied here - semi-supervised learning and clustering - seem quite a bit limited.
- While in principle the approach is applicable to various LVM problems the experiments only study two specific problems.
- The tasks considered here seem relatively simple, not enough to demonstrate the effect of the increased expressivity.
- The increased expressivity also comes at a computational cost.
- The method also introduces a critical new hyperparameter - the number of steps for diffusion. The paper does not discuss how this parameter is selected and how sensitive the results are to the choice of this parameter.
- Reproducibility: The authors do not include code to reproduce their results but most details seem to be included in the paper.
- While the regularization term in Eq. (3) is novel, the rest of the paper presents a standard hierarchical VAE.
- The method is compared to weak baselines.
- Section 3 is very dense, and could be structured better.
. An already dense paper that is hard to read.

Recommendation:

All reviewers lean towards rejection. I, therefore, recommend rejecting the paper and encourage the authors to use the feedback provided to improve the paper and resubmit to another venue.

**Justification For Why Not Higher Score:**

All reviewers vote for rejection. They point out numerous weaknesses.

**Justification For Why Not Lower Score:**

N/A

---

### Decision · Program_Chairs · 2024-01-16

Reject